# Increased listening effort and cochlear neural degeneration underlie speech-in-noise deficits in normal-hearing middle-aged adults

Maggie E Zink[1†], Leslie Zhen[1†], Jacie R McHaney[1†‡], Jennifer Klara[1], Kimberly Yurasits[1], Victoria E Cancel[1], Olivia Flemm[1], Claire Mitchell[1], Jyotishka Datta[2], Bharath Chandresekaran[1‡], Aravindakshan Parthasarathy[1,3,4]*

[1]Department of Communication Science and Disorders, School of Health and Rehabilitation Sciences, University of Pittsburgh, Pittsburgh, United States; [2]Department of Statistics, Virginia Polytechnic Institute and State University, Blacksburg, United States; [3]Department of Bioengineering, Swanson School of Engineering, University of Pittsburgh, Pittsburgh, United States; [4]Department of Otolaryngology, School of Medicine, University of Pittsburgh, Pittsburgh, United States

*For correspondence:
Aravind_Partha@pitt.edu

†These authors contributed equally to this work

Present address: ‡Department of Communication Sciences and Disorders, Northwestern University, Evanston, United States

Competing interest: The authors declare that no competing interests exist.

## eLife Assessment

This study aims to clarify the effects of cochlear neural degeneration on auditory processing in listeners with normal audiograms (sometimes referred to as 'hidden hearing loss'). The authors provide **important** new data demonstrating associations between cochlear neural degeneration, non-invasive assays of auditory processing, and speech perception. Based on a cross-species comparison, the findings pose **compelling** evidence that cochlear synaptopathy is associated with a significant part of hearing difficulties in complex environments.

**Abstract** Middle age represents a critical period of accelerated brain changes and provides a window for early detection and intervention in age-related neurological decline. Hearing loss is a key early marker of such decline and is linked to numerous comorbidities in older adults. Yet, ~10% of middle-aged individuals who report hearing difficulties show normal audiograms. Cochlear neural degeneration (CND) could contribute to these hidden hearing deficits, though its role remains unclear due to a lack of objective diagnostics and uncertainty regarding its perceptual outcomes. Here, we employed a cross-species design to examine neural and behavioral signatures of CND. We measured envelope following responses (EFRs) – neural ensemble responses to sound originating from the peripheral auditory pathway – in young and middle-aged adults with normal audiograms and compared these responses to young and middle-aged Mongolian gerbils, where CND was histologically confirmed. We observed near-identical changes in EFRs across species that were associated with CND. Behavioral assessments revealed age-related speech-in-noise deficits under challenging conditions, while pupil-indexed listening effort increased with age even when behavioral performance was matched. Together, these results demonstrate that CND contributes to speech perception difficulties and elevated listening effort in midlife, which may ultimately lead to listening fatigue and social withdrawal.

## Introduction

Age-related hearing loss, defined as declines in hearing sensitivity, is exceedingly common; according to some estimates, 45 million adults in the United States over 50 years of age have age-related hearing loss that is significant enough to interfere with communication (*Lin et al., 2011*). Untreated hearing loss decreases quality of life and is considered to be the single-largest modifiable risk factor in middle age for other age-related comorbidities such as cognitive impairment and dementia (*Livingston et al., 2017*). However, current measures of hearing sensitivity fail to capture critical aspects of real-world listening challenges in this population (*Hind et al., 2011*; *Tremblay et al., 2015*). Hearing difficulties experienced by up to 10% of adults seeking help in the hearing clinic are 'hidden' to current diagnostic procedures (*Hind et al., 2011*; *Tremblay et al., 2015*; *Cancel et al., 2023*; *Parthasarathy et al., 2020*). Peripheral deafferentation caused by cochlear neural degeneration (CND) may underlie many of these perceptual difficulties (*Kujawa and Liberman, 2009*; *Schaette and McAlpine, 2011*). Anatomical evidence for progressive CND with aging is clear – postmortem studies using human temporal bones estimate a 40% deafferentation caused by CND by the fifth decade of life (*Wu et al., 2019*; *Wu et al., 2023*; *Wu et al., 2020*). CND causes neural coding deficits in the peripheral auditory pathway, affecting the faithful representation of spectrotemporally complex auditory stimuli (*Parthasarathy and Kujawa, 2018*; *Mepani et al., 2021*; *Shaheen et al., 2015*). However, the evidence linking CND with perceptual deficits is mixed – current assessments of perceptual deficits associated with CND primarily focus on behavioral measures of speech in noise listening ability, with mixed evidence of deficits in individuals with putative CND (*Prendergast et al., 2017a*; *Guest et al., 2017*; *Grant et al., 2020*; *Grant et al., 2022*).

Two challenges impede our understanding of the perceptual consequences of CND. First, while many noninvasive markers of CND have been proposed and validated in animal models (*Kujawa and Liberman, 2009*; *Shaheen et al., 2015*; *Valero et al., 2018*; *Mehraei et al., 2016*), noninvasive estimates of putative CND in humans cannot be confirmed with histological assessment of synapses in the same participants. Cross-species comparative studies and computational modeling provide promising avenues for overcoming this gap (*Bharadwaj et al., 2022*; *Buran et al., 2022*). Second, behavioral readouts of perceptual difficulties in humans show mixed results, with putative CND depending on the specific test used and degree of spectrotemporal and contextual information provided in that test (*Grant et al., 2020*; *Prendergast et al., 2017b*; *Mepani et al., 2020*). The most promising tests for CND are ones with no linguistic context and short spectrotemporal processing windows (*Mepani et al., 2021*; *Mepani et al., 2020*). However, these behavioral readouts may minimize subliminal changes in perception that are reflected in listening effort but *not* in accuracies (*Pichora-Fuller et al., 2016*; *Peelle, 2018*; *Zekveld et al., 2011*). Specifically, two individuals may show similar accuracies on a listening task, but one individual may need to exert substantially more listening effort to achieve the same accuracy as the other. Here, we used a cross-species approach, combined with simultaneous measurements of behavior and listening effort, to show that CND was associated with decreased neural coding fidelity and increased listening effort in middle-aged adults with normal audiometric thresholds. We measured putative CND using the envelope following response (EFR) to rapid (~1000 Hz) modulation frequencies – a suggested marker for CND (*Parthasarathy and Kujawa, 2018*; *Shaheen et al., 2015*). Cross-species comparisons with identical recordings in a low-frequency hearing animal model, the Mongolian gerbil, confirmed that decreases in EFRs were selective only for responses with generators in the auditory nerve. These EFRs were also associated with histologically-confirmed CND in gerbils. In the human model, we simultaneously measured pupil-indexed listening effort in participants as they performed a speech-in-noise task and showed that increased listening effort was present despite matched behavioral accuracies. These results point to hitherto underexplored aspects of auditory perceptual difficulties associated with listening effort and CND.

## Results

### 'Normal' hearing middle-aged adults show evidence of peripheral neural coding deficits that are associated with CND

Middle-aged (40–55 years) and young adult (18–25 years) listeners were recruited to participate in this study (*Figure 1A*). All participants had clinically-normal hearing thresholds and spoke fluent American English. Participants had normal otoscopy by visual examination and air conduction thresholds

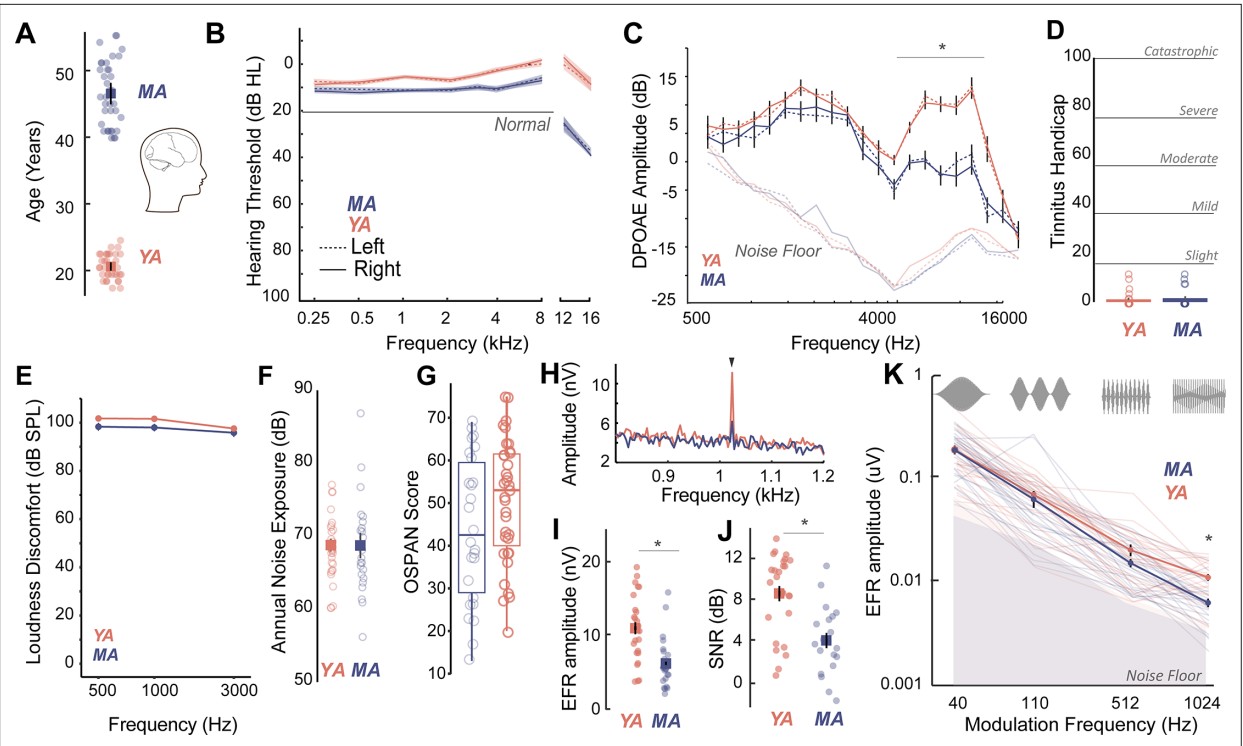

**Figure 1.** Age-related cochlear neural degeneration (CND) occurs prior to overt changes in hearing thresholds and can be assessed noninvasively by measuring phase-locked neural envelope following responses. (**A**) Thirty seven middle-aged (MA, 40–55 years, mean = 46.1 ± 4.6 years) and 35 young adults (YA, 18–25 years, mean = 21.17 ± 1.8 years) participated in this study. (**B**) All participants had clinically normal-hearing thresholds, with some evidence of threshold losses at extended high frequencies above 8 kHz typically not tested in the clinic. Hearing thresholds in dB HL are shown on the Y-axis and frequency in kHz is plotted on the X-axis. (**C**) Outer hair cell function assessed using distortion product otoacoustic emissions (DPOAEs) is comparable between YA and MA up to 4 kHz and showed age-related decreases at higher frequencies. Both cohorts show no evidence of self-reported tinnitus (**D**) or hyperacusis measured as loudness discomfort levels (LDLs) (**E**), have comparable self-reported noise exposure levels (**F**), and present with comparable working memory scores assessed using operation span task (OSPAN) (**G**). (**H**) Envelope following responses (EFRs) to modulation frequencies of 1024 Hz can be reliably recorded in YA and MA using 'tiptrodes'. The panel shows grand-averaged fast Fourier transform (FFT) traces for YA and MA. (**I**) MA showed significant declines in EFR amplitudes at 1024 Hz amplitude modulation (AM), with putative neural generators in the auditory nerve. (**J**) Signal-to-noise ratios were 8 dB on average for YA and 4 dB for MA. (**K**) Statistically significant decreases in EFR amplitudes were selective for 1024 Hz AM, the modulation frequency with putative generators in the auditory nerve. All panels: Error bars and shading represent standard error of the mean (SEM). Asterisks represent p<0.05, analysis of variance (ANOVA).

**Table 1.** Comparison of air conduction thresholds using a three-way analysis of variance (ANOVA) (middle-aged adults [MA] =37, young adults [YA] =35).

| Effects | DFn | Sum Sq | Mean Sq | F-value | *p*-Value |
|---|---|---|---|---|---|
| Frequency | 6 | 5067 | 844 | 20.786 | <0.001*** |
| Ear | 1 | 3 | 3 | 0.068 | 0.8 |
| Group | 1 | 5831 | 5813 | 143.521 | <0.001*** |
| Freq:Ear | 6 | 85 | 14 | 0.349 | 0.9 |
| Freq:Group | 6 | 905 | 151 | 3.712 | <0.001*** |
| Ear:Group | 1 | 7 | 7 | 0.164 | 0.7 |
| Freq:Ear:Group | 6 | 234 | 39 | 0.961 | 0.5 |
| Residuals | 840 | 34125 | 41 | | |

*p<0.05; ***p<0.001.

**Table 2.** Comparison of extended high frequencies using three-way analysis of variance (ANOVA) (middle-aged adults [MA] =37, young adults [YA] =35).

| Effects | DFn | Sum Sq | Mean Sq | F-value | *p*-Value |
|---|---|---|---|---|---|
| Frequency | 2 | 21209 | 10605 | 74.523 | <0.001*** |
| Ear | 1 | 6 | 6 | 0.039 | 0.8 |
| Group | 1 | 32868 | 32868 | 230.978 | <0.001*** |
| Freq:Ear | 2 | 142 | 142 | 0.498 | 0.6 |
| Freq:Group | 2 | 6016 | 6016 | 21.137 | <0.001*** |
| Ear:Group | 1 | 152 | 152 | 1.069 | 0.3 |
| Freq:Ear:Group | 2 | 38 | 19 | 0.134 | 0.9 |
| Residuals | 350 | 49805 | 142 | | |

*$p<0.05$; ***$p<0.001$.

≤25 dB HL for octave frequencies between 250 Hz and 8 kHz (*Figure 1B*, *Table 1*), consistent with WHO guidelines for normal hearing (*World Health Organization, 2024*). Threshold differences were exaggerated in MAs at extended high frequencies (>8 kHz), which are seldom clinically measured but may be a marker for accumulated lifetime noise damage (*Grant et al., 2020*; *Škerková et al., 2023*; *Mishra et al., 2022*; *Lough and Plack, 2022*, *Figure 1B*, *Table 2*). Outer hair cell function, assessed using distortion product otoacoustic emissions (DPOAEs), was comparable between young adult and middle-aged listeners up to 4 kHz, the frequency regions that contain most of the spectral information in speech (*Figure 1C*, *Table 3*). Participants also showed no severe symptoms of tinnitus (*Figure 1D*), assessed using the Tinnitus Handicap Inventory (THI; *Newman et al., 1996*) and loudness discomfort levels (LDLs; *Newman et al., 1996*) above 80 dB SPL for frequencies up to 3 kHz (*Figure 1E*, *Table 4*). Self-reported noise exposure using the Noise Exposure Questionnaire (NEQ; *Johnson et al., 2017*) was not significantly different between age groups (*Figure 1F*, *Table 4*). Participants also had normal cognitive function indexed by the Montreal Cognitive Assessment (MoCA>25; *Nasreddine et al., 2005*) and comparable working memory scores assessed using the operation span (OSPAN) task (*Turner and Engle, 1989*, *Figure 1G*, *Table 4*). Hence, the middle-aged adults recruited for this study were all considered 'normal' by currently administered behavioral and audiological assessments in the hearing clinic, while exhibiting some subclinical outer hair cell dysfunction, especially at frequencies above 4 kHz.

We then measured putative CND using neural ensemble responses from the auditory periphery phase-locked to the stimulus amplitude envelope via the EFR. EFRs can be used to emphasize neural generators in the auditory periphery by exploiting divergent phase-locking abilities along the ascending auditory pathway. EFRs at rapid amplitude modulation (AM) frequencies above 600 Hz have been shown to relate to underlying CND in animal models (*Parthasarathy and Kujawa, 2018*; *Shaheen et al., 2015*) and in humans (*Mepani et al., 2021*). Here, we measured EFRs to AM frequencies that have putative neural generators in the central auditory pathway such as the cortex (40 Hz AM) (*Parthasarathy and Kujawa, 2018*; *Parthasarathy and Bartlett, 2012*), as well as faster modulation rates (110 Hz, 512 Hz, and 1024 Hz AM) that emphasize progressively peripheral auditory regions (*Parthasarathy and Kujawa, 2018*). We were able to reliably record EFRs up to 1024 Hz by using gold-foil tipped electrodes ('tiptrodes') placed in the ear canal, closer to the presumptive neural generators in the auditory nerve (*Figure 1H*). EFR peaks analyzed in the spectral domain were above the noise floor, with average signal-to-noise ratios (SNRs) of 8 dB in younger and 4 dB in middle-aged adults (the panels I and J of *Figure 1*). Statistically significant age-related decreases in EFR amplitudes were only present for EFRs to the 1024 Hz AM rate, which has putative generators in the auditory nerve (*Parthasarathy and Kujawa, 2018*; *Shaheen et al., 2015*), but were not present for slower AM rates with putative generators in the midbrain or cortex (*Figure 1K*, *Table 5*).

To confirm that the EFR parameters used here were indeed sensitive to putative CND, we measured EFRs using identical stimuli, acquisition, and analysis parameters in young (22 weeks) and middle-aged (80 weeks) Mongolian gerbils (*Figure 2A*). The hearing range of gerbils largely overlaps

**Table 3.** Comparison of right ear distortion product otoacoustic emissions using a two-way analysis of variance (ANOVA) (middle-aged adults [MA] =34, young adults [YA] =31).

| Effects | DFn | DFd | F-value | p-Value |
|---|---|---|---|---|
| Group | 1 | 63 | 25.85 | <0.001*** |
| Freq | 9.55 | 601.56 | 58.786 | <0.001*** |
| Group:Freq | 9.55 | 601.56 | 7.341 | <0.001*** |
| 501:Group | 1 | 63 | 0.713 | 1.00 |
| 595:Group | 1 | 63 | 1.939 | 1.00 |
| 707:Group | 1 | 63 | 0.718 | 1.00 |
| 841:Group | 1 | 63 | 0.268 | 1.00 |
| 998:Group | 1 | 63 | 4.38 | 0.84 |
| 1188:Group | 1 | 63 | 0.794 | 1.00 |
| 1414:Group | 1 | 63 | 4.67 | 0.74 |
| 1681:Group | 1 | 63 | 1.724 | 1.00 |
| 2000:Group | 1 | 63 | 0.87 | 1.00 |
| 2378:Group | 1 | 63 | 0.059 | 1.00 |
| 2828:Group | 1 | 63 | 4.755 | 0.69 |
| 3365:Group | 1 | 63 | 2.095 | 1.00 |
| 4001:Group | 1 | 63 | 10.463 | 0.04* |
| 4757:Group | 1 | 63 | 18.015 | <0.001*** |
| 5658:Group | 1 | 63 | 29.947 | <0.001*** |
| 6727:Group | 1 | 63 | 37.01 | <0.001*** |
| 8000:Group | 1 | 63 | 28.94 | <0.001*** |
| 9514:Group | 1 | 63 | 39.235 | <0.001*** |
| 11314:Group | 1 | 63 | 26.847 | <0.001*** |
| 13454:Group | 1 | 63 | 7.771 | 0.147 |
| 160000:Group | 1 | 63 | 0.436 | 1.00 |

*$p < .05$; ***$p < .001$.

with that of humans at speech frequencies (**Ryan, 1976**), making them an ideal animal model for direct comparison in cross-species studies. Middle-aged gerbils showed no loss of hearing thresholds, similar to middle-aged humans (**Figure 2B**). Remarkably, gerbils also exhibited a selective decrease in EFR amplitudes for AM rates at 1024 Hz, similar to middle-aged humans (**Figure 2C**, **Table 6**). CND in gerbils was assessed using immunohistological analysis of cochlear whole mounts where the cell bodies, presynaptic ribbon terminals, and the postsynaptic glutamate receptor patches were

**Table 4.** Comparisons using one-way analyses of variance (ANOVAs).

| Measure | YA (n) | MA (n) | DFn | DFd | F-value | p-Value |
|---|---|---|---|---|---|---|
| THI | 33 | 37 | 1 | 68 | 0.834 | 0.364 |
| OSPAN | 34 | 34 | 1 | 66 | 3.501 | 0.066 |
| QuickSIN Clinical Score | 31 | 34 | 1 | 63 | 3.214 | 0.078 |
| NEQ | 32 | 32 | 1 | 66 | 0.8375 | 0.363 |

Adjusted *p*-values are reported using Bonferroni correction.

**Table 5.** Comparison of envelope following responses (EFRs) using two-way analyses of variance (ANOVAs) (middle-aged adults [MA] =29, young adults [YA] =28).

| Effects | DFn | DFd | F-value | *p*-Value |
|---|---|---|---|---|
| Group | 1 | 54 | 0.275 | 0.6 |
| AM | 1.47 | 79.49 | 151.407 | <0.001*** |
| Group:AM | 1.47 | 79.49 | 0.151 | 0.929 |
| 1024:Group | 1 | 55 | 23.8 | <0.001*** |
| 512:Group | 1 | 56 | 3.171 | 0.083 |
| 110:Group | 1 | 55 | 0.491 | 0.487 |
| 40:Group | 1 | 54 | 0.027 | 0.870 |

*$p<0.05$; ***$p<0.001$.

immunostained, visualized using confocal microscopy, and quantified from 3D reconstructed images (*Figure 2D*). Significant decreases in afferent synapse counts were present in middle-aged gerbils, reaching up to 20% losses compared to the young gerbils (*Figure 2E*, *Table 7*). Further, EFR amplitudes were significantly correlated to the number of remaining cochlear synapses (*Figure 2F*), thus confirming that our EFRs were a sensitive metric of CND.

## Perceptual deficits manifest as increased listening effort prior to behavioral deficits in middle-aged adults

Do middle-aged adults with putative CND experience challenges with hearing in noise despite having clinically normal-hearing thresholds? We measured speech perception in noise abilities with the clinically-used Quick Speech-in-Noise (QuickSIN; *Killion et al., 2004*) task to assess hearing in noise changes that mimic real-world listening scenarios. The QuickSIN tests suprathreshold hearing of medium-context sentences presented in varying levels of four-talker background babble ranging from 25 dB to 0 dB SNR levels in 5 dB steps (*Figure 3A*). Further, QuickSIN is a clinically-relevant test that we recently identified as being sensitive to perceptual deficits in adult populations with normal audiograms (*Cancel et al., 2023*). On each trial, participants were instructed to repeat a target sentence, which contained five keywords for identification. Clinically, QuickSIN is scored as dB SNR loss, reflecting the SNR level required to accurately identify keywords in noise correctly half the

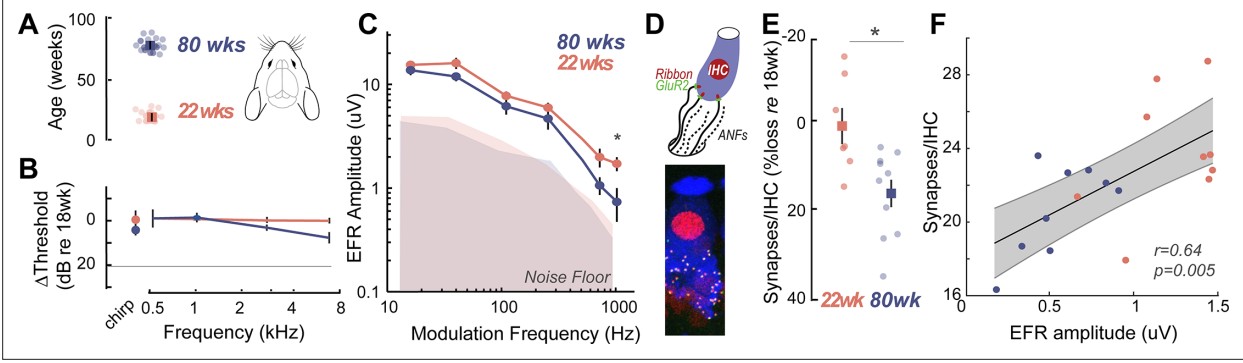

**Figure 2.** Cross-species experiments in a rodent model show that envelope following responses (EFRs) are a sensitive biomarker for histologically-confirmed cochlear neural degeneration (CND). (**A**) Cross-species comparisons were made with young (22±0.86 weeks, *n*=14) and middle-aged (80±0.76 weeks, *n*=12) Mongolian gerbils, with identical stimuli, recording, and analysis parameters. (**B**) Middle-aged gerbils did not show any age-related decreases in hearing thresholds. (**C**) Age-related decreases in EFR amplitudes were isolated to the 1024 Hz modulation frequency, similar to middle-aged humans in *Figure 1K*. (**D**) CND was quantified using immunostained organ of Corti whole mounts, where afferent excitatory synapses were quantified using 3D reconstructed images. (**E**) Cochlear synapse counts at the 3 kHz cochlear region corresponding to the carrier frequency for the EFRs were significantly decreased in middle-aged gerbils, despite matched auditory thresholds. (**F**) EFR amplitudes at 1024 Hz amplitude modulation (AM) were significantly correlated with the number of remaining cochlear synapses, suggesting that these EFRs are a sensitive metric for CND with age. All panels: Error bars and shading represent standard error of the mean (SEM). Asterisks represent p<0.05, analysis of variance (ANOVA).

**Table 6.** Comparison of 22-week-old gerbil (*n*=14) and 80-week-old gerbil (*n*=12) envelope following responses (EFRs) using two-way analyses of variance (ANOVAs).

| Effects | DFn | DFd | F-value | *p*-Value |
|---|---|---|---|---|
| Group | 1 | 24 | 4.125 | 0.053 |
| AM | 2.68 | 64.28 | 74.636 | <0.001*** |
| Group:AM | 2.68 | 64.28 | 0.875 | 0.449 |
| 16:Group | 1 | 24 | 0.456 | 0.506 |
| 40:Group | 1 | 24 | 2.461 | 0.130 |
| 110:Group | 1 | 24 | 3.056 | 0.093 |
| 256:Group | 1 | 24 | 1.959 | 0.174 |
| 724:Group | 1 | 24 | 2.483 | 0.128 |
| 1024:Group | 1 | 24 | 5.158 | 0.032* |

*$p < .05$; ***$p < .001$.

time. No significant age-related differences were observed in clinically scored QuickSIN dB SNR loss (*Figure 3B*, *Table 4*). When analyzing performance at each SNR, accuracy was at near-ceiling from 25 dB SNR to 10 dB SNR, but dropped from 5 dB SNR in both young and middle-aged adults. Statistically significant behavioral deficits with age were observed on QuickSIN only in the most challenging SNR of 0 dB (*Figure 3C*, *Table 8*).

Are there perceptual deficits experienced by middle-aged adults that are not captured by traditional behavioral readouts? We addressed this question by measuring isoluminous task-related changes in pupil diameter as an index of listening effort (*Beatty, 1982*; *Winn et al., 2015*; *Kuchinsky et al., 2013*) while participants performed the QuickSIN task (*Figure 3A*). Pupillary changes were analyzed using growth curve analysis (GCA, *Mirman, 2014*). GCAs provide a statistical approach to modeling changes over time in the timing and shape of the pupillary response and have several advantages to analyzing pupillary response over traditional approaches. First, GCA does not require time-binned samples, thus removing the trade-off between temporal resolution and statistical power, and, second, GCA can account for individual variability. Two second-order GCAs were fit to different time-windows (*Tables 9–10*, see Methods). One time-window encompassed the onset of the masker through the first 2.8 s of the target sentence (listening window). The second window spanned from the offset of the target sentence up to the verbal response prompt (integration window). These two time-windows were hypothesized to represent effort associated with differing sensory and cognitive processes. The listening window reflects linguistic and semantic processing of ongoing speech stimuli and is a physiological response to auditory processing (*McHaney et al., 2021*), while the integration window reflects error correction, working memory, and comparisons with predictive internal models (*Kuchinsky et al., 2014*; *Winn, 2023*). The linear term from the GCA was further analyzed as a marker for the slope of pupillary change over time.

Pupil-indexed listening effort measured during listening was modulated by task difficulty, with pupil diameters showing a larger increase at more challenging SNRs (*Figure 3D*). Both younger and middle-aged adults showed increases in pupil-indexed effort prior to overt decreases in behavioral performance (*Figure 3E*). While MAs exhibited larger increases in listening effort compared to YAs, this change was not statistically significant (*Figure 3E*, *Table 9*). Trends seen in the pupillary responses for the listening window were further amplified in the integration window (*Figure 3F*). Pupillary slopes obtained from the GCA increased with task difficulty for both YA and MA. However, middle-aged

**Table 7.** Comparison of synapse counts at 3000 Hz in 22- and 80-week-old gerbils using one-way analysis of variance (ANOVA).

| Measure | 22 weeks (*n*) | 80 weeks (*n*) | DFn | DFd | F-value | *p*-Value |
|---|---|---|---|---|---|---|
| Synapse counts | 14 | 12 | 1 | 16 | 4.877 | 0.042* |

*$p<0.05$; ***$p<0.001$.

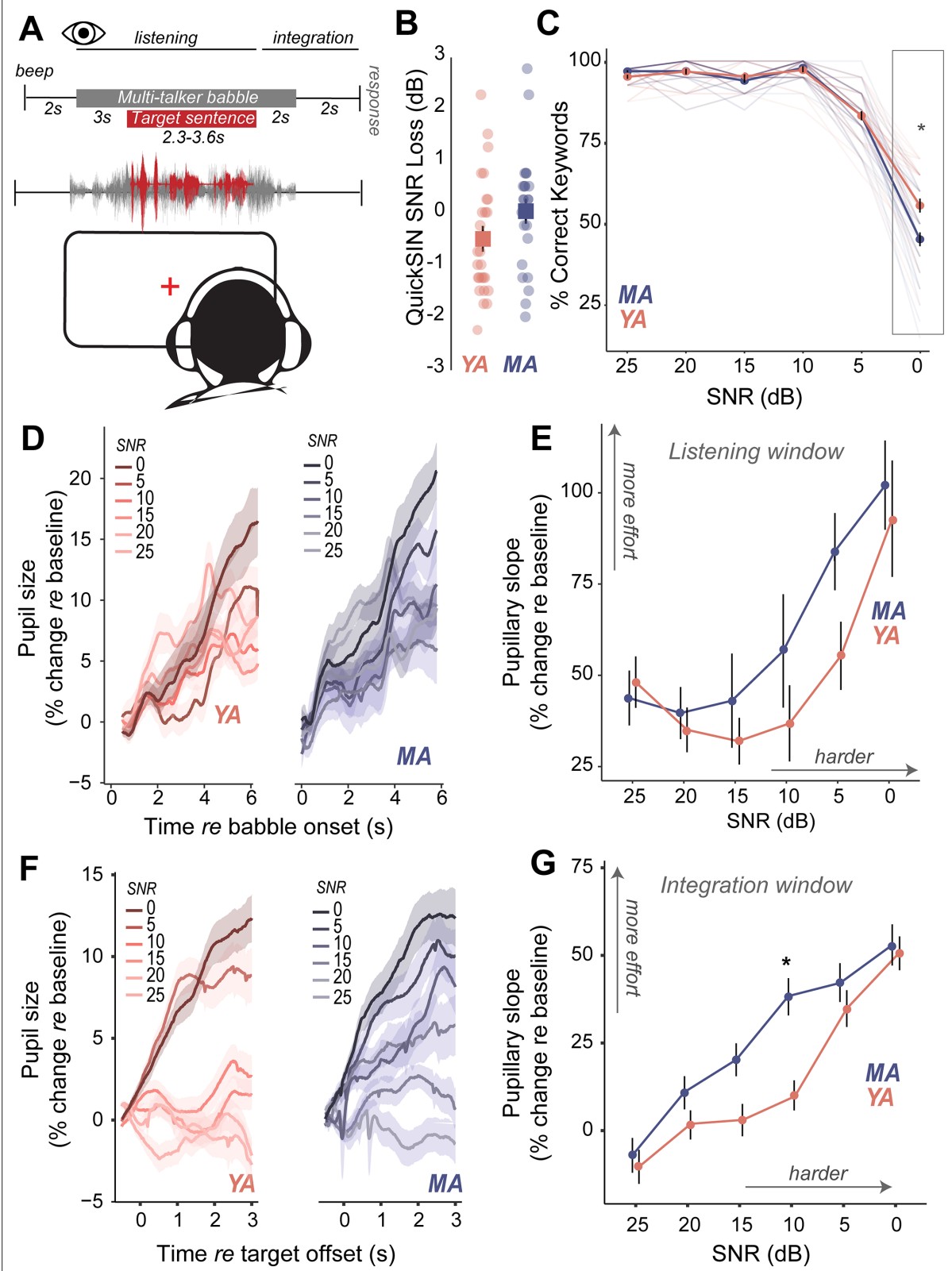

**Figure 3.** Increased listening effort precedes behavioral deficits in speech-in-noise perception in middle-aged adults. (**A**) Speech perception in noise was assessed using the Quick Speech-in-Noise (QuickSIN) test, which presents moderate context sentences in varying levels of multi-talker babble. Pupillary measures were analyzed in two time-windows: (1) during stimulus presentation and (2) after target sentence offset and prior to response initiation. (**B**) No significant age-related differences were observed in clinical QuickSIN scores presented as dB signal-to-noise ratio (SNR) loss. (**C**)

*Figure 3 continued on next page*

*Figure 3 continued*

QuickSIN performance is matched between middle-aged (MA) and younger adults (YA) until the most difficult noise condition (SNR 0). The x-axis shows the SNR condition that the target sentences were presented in, with 25 dB being the easiest noise condition, and 0 dB being the most difficult noise condition. The y-axis shows participant accuracy in repeating keywords from the target sentences as percent correct. (**D**) Grand-averaged pupillary responses, measured during task listening as an index of effort, exhibit modulation with task difficulty, with greater pupillary dilations observed in harder conditions for both groups. (**E**) Middle-aged adults show consistently higher pupillary responses during performance on the QuickSIN task and at SNR levels prior to when overt behavioral deficits are observed. (**F**) Grand-averaged pupillary responses measured after target sentence offset as an index of effort exhibit greater modulation with task difficulty, compared to changes in the listening window. (**G**) Trends seen in the listening window were amplified in this integration window, with middle-aged adults showing even greater effort, especially at moderate SNRs where behavior was matched.

adults showed a larger increase in listening effort than younger adults with decreasing SNRs, with significant age group listening effort differences at 10 dB SNR, even though behavioral performance was matched (*Figure 3G*, *Table 10*). These results suggest that middle-aged adults may maintain comparable performance to younger listeners at moderate task difficulty, but at the cost of recruiting listening effort.

## Pupil-indexed listening effort and CND provide synergistic contributions to speech-in-noise intelligibility

We sought to understand the relationships between CND, listening effort, and speech-in-noise intelligibility in normal-hearing, middle-aged adults. Behavioral performance in QuickSIN at 0 dB SNR, where there was a group effect of age, was significantly correlated with putative CND assessed using EFRs at 1024 Hz (*Figure 4A*). This suggests that peripheral deafferentation may manifest as overt behavioral deficits under the most challenging listening conditions. Pupil-indexed listening effort was also greater in the integration window in middle-aged adults at 10 dB SNR compared to younger adults (*Figure 3G*), even though behavioral performance was near-ceiling for both age groups. Pupillary slopes at 10 dB SNR in the integration window were correlated with behavioral deficits at 0 dB SNR (*Figure 4B*). These results add to the growing evidence, suggesting that pupil-indexed listening effort to maintain behavioral performance at moderate task difficulties is predictive of behavioral performance at more challenging listening conditions (*McHaney et al., 2024*). There were significant correlations between pupillary slopes in the listening window as well, even though there were no group-level differences with age (*Figure 4C*). These data suggest that CND and increased listening effort are both associated with listening challenges in middle-aged adults.

Is the increase in listening effort synergistic with CND? To understand the multifactorial contributions of sensory and top-down factors that may affect speech perception in noise, we performed a penalized regression with elastic net penalty (*Zou and Hastie, 2005*). QuickSIN performance at 0 dB SNR (scaled to 0–100) was used as the outcome variable and all other measured variables were inserted as input variables. The elastic net penalized regression framework is a robust method that

**Table 8.** Comparison of Quick Speech-in-Noise (QuickSIN) performance using a mixed-design analysis of variance (ANOVA) (middle-age adults [MA] =34, young adults [YA] =31).

| Effects | DFn | DFd | F-value | *p*-value |
|---|---|---|---|---|
| Group | 1 | 372 | 3.79 | 0.0522 |
| SNR | 5 | 372 | 541.81 | <0.001*** |
| Group:SNR | 5 | 372 | 6.57 | <0.001*** |
| 0:Group | 1 | 62 | 10.512 | 0.001* |
| 5:Group | 1 | 62 | 0.002 | 0.956 |
| 10:Group | 1 | 62 | 1.336 | 0.252 |
| 15:Group | 1 | 62 | 1.834 | 0.180 |
| 20:Group | 1 | 62 | 0.227 | 0.634 |
| 25:Group | 1 | 62 | 4.980 | 0.0292* |

*$p < .05$; ***$p < .001$.

**Table 9.** Fixed-effect estimates for the model of pupillary responses from 0 s to 5.8 s time-locked to the babble masker onset to examine the effect of signal-to-noise ratio (SNR) and age group (observations =96,612, groups: participant × SNR =332, participant =63).

| Fixed effect | Estimate | SE | 95% CI | t | p |
|---|---|---|---|---|---|
| Intercept | 7.26 | 1.26 | [4.79, 9.73] | 5.76 | <0.001*** |
| ot1 | 48.14 | 10.62 | [27.33, 68.94] | 4.53 | <0.001*** |
| ot2 | −30.44 | 6.90 | [−43.96,−16.92] | −4.41 | <0.001*** |
| SNR 0 | −0.43 | 1.80 | [−3.95, 3.09] | −0.24 | 0.811 |
| SNR 5 | −3.45 | 1.80 | [−6.98, 0.07] | −1.92 | 0.055 |
| SNR 10 | −3.53 | 1.80 | [−7.06,−0.01] | −1.96 | 0.049* |
| SNR 15 | −3.72 | 1.77 | [−7.19,−0.25] | −2.10 | 0.035* |
| SNR 20 | −2.32 | 1.80 | [−5.85, 1.20] | −1.29 | 0.197 |
| ot1 × SNR 0 | 44.77 | 14.58 | [16.19, 73.34] | 3.07 | 0.002* |
| ot1 × SNR 5 | 7.21 | 14.59 | [−21.39, 35.81] | 0.49 | 0.621 |
| ot1 × SNR 10 | −11.30 | 14.59 | [−39.90, 17.30] | −0.77 | 0.439 |
| ot1 × SNR 15 | −16.19 | 14.34 | [−44.29, 11.92] | −1.13 | 0.259 |
| ot1 × SNR 20 | −13.08 | 14.58 | [−41.65, 15.50] | −0.90 | 0.370 |
| ot2 × SNR 0 | 44.88 | 9.26 | [26.73, 63.02] | 4.85 | <0.001*** |
| ot2 × SNR 5 | 61.77 | 9.27 | [43.60, 79.94] | 6.66 | <0.001*** |
| ot2 × SNR 10 | 26.06 | 9.27 | [7.88, 44.23] | 2.81 | 0.005* |
| ot2 × SNR 15 | 14.74 | 9.11 | [−3.11, 32.59] | 1.62 | 0.105 |
| ot2 × SNR 20 | 20.07 | 9.26 | [1.92, 38.22] | 2.17 | 0.030* |
| Group (MA vs. YA) | 1.03 | 1.85 | [−2.60, 4.65] | 0.55 | 0.579 |
| Group × SNR 0 | 1.53 | 2.64 | [−3.65, 6.70] | 0.58 | 0.564 |
| Group × SNR 5 | 1.87 | 2.63 | [−3.28, 7.02] | 0.71 | 0.476 |
| Group × SNR 10 | −4.82e−03 | 2.60 | [−5.10, 5.10] | −1.85e−03 | 0.999 |
| Group × SNR 15 | −1.26 | 2.62 | [−6.39, 3.88] | −0.48 | 0.632 |
| Group × SNR 20 | −1.26 | 2.64 | [−6.44, 3.91] | −0.48 | 0.632 |
| ot1 × Group | −4.37 | 15.56 | [−34.87, 26.14] | −0.28 | 0.779 |
| ot1 × Group × SNR 0 | 13.71 | 21.46 | [−28.35, 55.77] | 0.64 | 0.523 |
| ot1 × Group × SNR 5 | 32.89 | 21.35 | [−8.94, 74.73] | 1.54 | 0.123 |
| ot1 × Group × SNR 10 | 24.20 | 21.16 | [−17.26, 65.66] | 1.14 | 0.253 |
| ot1 × Group × SNR 15 | 15.46 | 21.28 | [−26.25, 57.18] | 0.73 | 0.468 |
| ot1 × Group × SNR 20 | 8.97 | 21.45 | [−33.07, 51.00] | 0.42 | 0.676 |
| ot2 × Group | −4.89 | 10.11 | [−24.70, 14.92] | −0.48 | 0.628 |
| ot2 × Group × SNR 0 | 3.18 | 13.65 | [−23.57, 29.93] | 0.23 | 0.816 |
| ot2 × Group × SNR 5 | −11.65 | 13.57 | [−38.26, 14.96] | −0.86 | 0.391 |
| ot2 × Group × SNR 10 | 16.76 | 13.46 | [−9.62, 43.13] | 1.24 | 0.213 |
| ot2 × Group × SNR 15 | 14.82 | 13.53 | [−11.70, 41.35] | 1.10 | 0.273 |
| ot2 × Group × SNR 20 | 12.19 | 13.63 | [−14.54, 38.91] | 0.89 | 0.371 |

Growth curve formula: lmer(Pupil ~ (ot1 + ot2)*Group*SNR + (0+ot1+ot2 | participant) + (ot1 +ot2 | participant:SNR), control = lmerControl(optimizer = 'bobyqa'), REML = FALSE). Orthogonal polynomial terms: ot1=linear (slope); ot2=quadratic (curvature).

*p<0.05; ***p<0.001.

**Table 10.** Fixed-effect estimates for model of pupillary responses from 0 s to 3 s time-locked to Quick Speech-in-Noise (QuickSIN) target sentence offset to examine the effect of signal-to-noise ratio (SNR) and age group (observations =63,184, groups: participant × SNR =359, participant =63).

| Fixed effect | Estimate | SE | 95% CI | t | p |
|---|---|---|---|---|---|
| Intercept | –0.36 | 0.81 | [–1.95, 1.22] | –0.45 | 0.652 |
| ot1 | –10.33 | 6.06 | [–22.20, 1.54] | –1.71 | 0.088 |
| ot2 | –2.24 | 3.12 | [–8.35, 3.88] | –0.72 | 0.474 |
| SNR 0 | 7.40 | 1.00 | [5.45, 9.36] | 7.43 | <0.001*** |
| SNR 5 | 6.93 | 1.00 | [4.97, 8.88] | 6.95 | <0.001*** |
| SNR 10 | 1.86 | 1.00 | [–0.09, 3.82] | 1.87 | 0.062 |
| SNR 15 | 0.84 | 1.01 | [–1.13, 2.81] | 0.83 | 0.404 |
| SNR 20 | –0.55 | 1.00 | [–2.50, 1.41] | –0.55 | 0.583 |
| ot1 × SNR 0 | 60.92 | 7.15 | [46.91, 74.92] | 8.52 | <0.001*** |
| ot1 × SNR 5 | 45.16 | 7.15 | [31.15, 59.16] | 6.32 | <0.001*** |
| ot1 × SNR 10 | 20.10 | 7.15 | [6.10, 34.11] | 2.81 | 0.005* |
| ot1 × SNR 15 | 13.38 | 7.21 | [–0.76, 27.51] | 1.85 | 0.064 |
| ot1 × SNR 20 | 12.27 | 7.15 | [–1.74, 26.28] | 1.72 | 0.086 |
| ot2 × SNR 0 | –3.41 | 4.19 | [–11.62, 4.81] | –0.81 | 0.416 |
| ot2 × SNR 5 | –14.97 | 4.19 | [-23.19,–6.75] | –3.57 | <0.001*** |
| ot2 × SNR 10 | 6.43 | 4.19 | [–1.78, 14.65] | 1.53 | 0.125 |
| ot2 × SNR 15 | 8.83 | 4.23 | [0.54, 17.12] | 2.09 | 0.037* |
| ot2 × SNR 20 | 7.83 | 4.19 | [–0.39, 16.05] | 1.87 | 0.062 |
| Group (MA vs YA) | –0.30 | 1.16 | [–2.57, 1.97] | –0.26 | 0.796 |
| Group × SNR 0 | 1.64 | 1.44 | [–1.18, 4.46] | 1.14 | 0.254 |
| Group × SNR 5 | 0.37 | 1.43 | [–2.43, 3.16] | 0.26 | 0.796 |
| Group × SNR 10 | 3.16 | 1.43 | [0.36, 5.97] | 2.21 | 0.027* |
| Group × SNR 15 | 3.79 | 1.45 | [0.95, 6.63] | 2.62 | 0.009* |
| Group × SNR 20 | 2.63 | 1.45 | [–0.22, 5.47] | 1.81 | 0.071 |
| ot1 × Group | 3.28 | 8.67 | [–13.72, 20.27] | 0.38 | 0.706 |
| ot1 × Group × SNR 0 | –0.89 | 10.33 | [–21.13, 19.36] | –0.09 | 0.932 |
| ot1 × Group × SNR 5 | 4.05 | 10.23 | [–15.99, 24.10] | 0.40 | 0.692 |
| ot1 × Group × SNR 10 | 25.33 | 10.26 | [5.21, 45.44] | 2.47 | 0.014* |
| ot1 × Group × SNR 15 | 14.01 | 10.40 | [–6.37, 34.39] | 1.35 | 0.178 |
| ot1 × Group × SNR 20 | 6.24 | 10.43 | [–14.20, 26.67] | 0.60 | 0.550 |
| ot2 × Group | 5.50 | 4.48 | [–3.29, 14.29] | 1.23 | 0.220 |
| ot2 × Group × SNR 0 | –11.67 | 6.04 | [–23.51, 0.18] | –1.93 | 0.053 |
| ot2 × Group × SNR 5 | 3.62 | 5.99 | [–8.11, 15.36] | 0.61 | 0.545 |
| ot2 × Group × SNR 10 | –6.72 | 6.01 | [–18.50, 5.06] | –1.12 | 0.264 |
| ot2 × Group × SNR 15 | –18.83 | 6.09 | [-30.77,–6.90] | –3.09 | 0.002* |
| ot2 × Group × SNR 20 | –17.10 | 6.10 | [-29.06,–5.15] | –2.80 | 0.005* |

Growth curve formula: *lmer(Pupil ~ (ot1 +ot2)\*Group\*SNR + (ot1 +ot2 | participant) + (ot1 + ot2 | participant:SNR), control = lmerControl(optimizer = 'bobyqa'), REML = FALSE)*.
Orthogonal polynomial terms: ot1=linear (slope); ot2=quadratic (curvature).
*p<0.05; ***p<0.001.

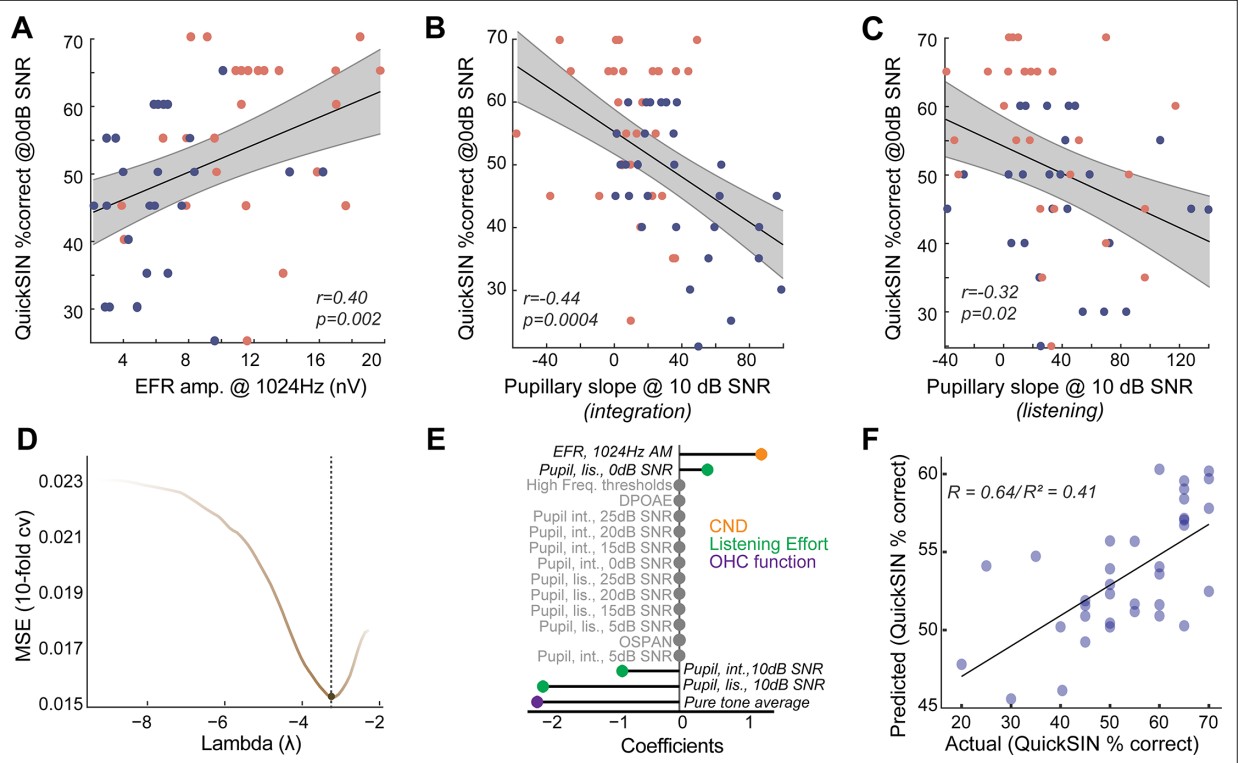

**Figure 4.** Listening effort and cochlear neural degeneration (CND) provide complementary contributions to speech-in-noise intelligibility. (**A**) Behavioral performance at the most challenging signal-to-noise ratio (SNR) was significantly correlated with the envelope following response (EFR) measures of CND, with lower EFR amplitudes being associated with poorer behavioral performance. (**B**) Pupillary responses at 10 dB SNR from the integration window were significantly correlated with behavioral performance at 0 dB SNR. (**B**) These correlations between pupillary responses at 10 dB SNR and behavioral performance at 0 dB SNR were also found in the listening window, even though there were no group differences in age, further strengthening the link between listening effort at moderate SNRs and behavioral performance at challenging SNRs. (**D**) An elastic net regression model with 10-fold cross-validation (cv) was fit to the Quick Speech-in-Noise (QuickSIN) scores at 0 dB SNR. The tuning parameter Lambda controls the extent to which coefficients contributing least to predictive accuracy are suppressed. (**E**) A lollipop plot displaying the coefficients ($\beta$) contributing to explaining variance on QuickSIN performance suggests that CND, listening effort, and subclinical changes in hearing thresholds all contribute to QuickSIN performance. (**F**) QuickSIN scores predicted by the elastic net regression are correlated with actual participant QuickSIN scores.

blends Lasso's ability to perform variable selection and Ridge's ability to handle multicollinearity and grouped covariates. The fitted elastic net regression model showed an $R^2$ value of 0.5981 and five significant predictors – hearing thresholds averaged across 500 Hz to 4 kHz (PTA4k), EFR amplitudes at 1024 Hz AM, pupillary slopes at 10 dB SNR and 0 dB SNR in the listening window, and pupillary slopes at 10 dB SNR in the integration window (*Figure 4D and E*). This model was significantly related to QuickSIN performance and predicted the observed QuickSIN scores across younger and middle-aged adults (r=0.64/(pseudo-)$R^2$=0.41, *Figure 4F*). Hence, the output of the elastic net regression suggests that CND and pupil-indexed listening, in addition to subclinical changes in hearing thresholds, all provided complementary contributions to speech perception in noise.

## Discussion

Middle age, typically defined as the fifth and sixth decade of life, has been historically understudied compared to older age ranges (*Dohm-Hansen et al., 2024*). Increasing evidence suggests that middle age is a critical period of rapid changes in brain function (*Schaum et al., 2020*; *Salthouse, 2019*). The resilience of the brain in keeping with degenerative processes that begin to occur in middle age predicts further age-related degeneration in later life and presents a critical opportunity for early intervention (*Dohm-Hansen et al., 2024*; *Elliott et al., 2021a*; *Elliott et al., 2021b*; *Hughes et al., 2018*). Hearing loss in middle age has recently been identified as the largest modifiable risk factor for dementia and Alzheimer's disease later in life (*Livingston et al., 2017*). However, the number of

middle-aged patients who seek help for hearing difficulties but show no abnormal clinical indicators suggests the need for the development of sensitive biomarkers for hearing challenges experienced by this population (*Hind et al., 2011*; *Cancel et al., 2023*; *Parthasarathy et al., 2020*; *Spehar and Lichtenhan, 2018*).

Anatomical evidence from human temporal bones suggests a 40% deafferentation of cochlear synapses in middle-aged adults, even without substantial noise exposure history (*Wu et al., 2019*; *Wu et al., 2023*; *Wu et al., 2020*). Peripheral deafferentation triggers compensatory mechanisms across sensory, language, and attentional systems (*Auerbach et al., 2019*; *Chambers et al., 2016*; *Resnik and Polley, 2021*; *Bharadwaj et al., 2015*). But our understanding of the perceptual consequences of cochlear deafferentation is limited by the lack of consensus on sensitive biomarkers for CND (*Bramhall et al., 2019*). Recent studies have identified multiple promising biomarkers for CND in animal models and human populations (*Bharadwaj et al., 2022*; *Mepani et al., 2021*; *Bharadwaj et al., 2019*). Reduced wave I amplitudes in the auditory brainstem response are a reliable marker of CND in animal models (*Kujawa and Liberman, 2009*; *Parthasarathy and Kujawa, 2018*; *Sergeyenko et al., 2013*), but can be challenging to obtain in humans (*Mehraei et al., 2016*; *Bharadwaj et al., 2019*). The middle-ear muscle reflex, an acoustic measurement of middle-ear immittance driven by efferent feedback to the middle-ear muscles, has also been identified as a promising marker for CND (*Valero et al., 2018*; *Bharadwaj et al., 2022*; *Valero et al., 2016*). Here, we used the EFR to identify CND in middle-aged adults with normal audiometric thresholds. As opposed to the middle-ear muscle reflex, EFRs measure peripheral neural coding and central auditory activity by exploiting the divergent phase-locking abilities of the ascending auditory pathway (*Joris et al., 2004*; *Parida et al., 2024*). EFRs with modulation rates greater than ~1000 Hz have been associated with CND and are considered to reflect the integrity of the auditory nerve (*Parthasarathy and Kujawa, 2018*; *Shaheen et al., 2015*), given that midbrain and cortical neurons cannot phase-lock to such high rates (*Joris et al., 2004*). We observed decreases in EFRs at modulation rates that were selective to the auditory periphery (i.e. 1024 Hz) in middle-aged adults, while EFRs at slower modulation rates, likely generated from the central auditory structures, were not different from those in younger adults (*Figure 1K*). The use of a more rapid onset time in the stimulus modulation envelope, such as the rectangular amplitude modulated tones (RAM EFRs), may result in a larger separation of these groups, even at slower modulation rates (*Vasilkov et al., 2021*; *Garrett et al., 2025*), as sharper onset times result in greater EFR amplitudes (*Mepani et al., 2021*; *Parthasarathy and Bartlett, 2011*). However, a more intriguing possibility is that middle-aged adults exhibited an increase in relative central auditory activity, or 'gain', in the presence of decreased peripheral neural coding (*Auerbach et al., 2019*; *Resnik and Polley, 2021*). The perceptual consequences of this gain are unclear, but our findings align with emerging evidence, suggesting that gain is associated with selective deficits in speech-in-noise abilities (*Resnik and Polley, 2021*; *Dougherty et al., 2021*; *Rumschlag et al., 2022*). EFRs at suprathreshold levels presented here also have contributions from higher-frequency regions due to a broader excitation at the cochlea (*Parthasarathy et al., 2016*; *Lai and Bartlett, 2018*). Since cochlear synapse loss is also believed to be flat across frequencies with age, EFRs used here likely index cochlear synapse loss equally across a broad range of frequencies (*Wu et al., 2019*; *Parthasarathy and Kujawa, 2018*; *Sergeyenko et al., 2013*). This notion is further supported by emerging evidence that suggests that phase-locking measured to lower frequency pure tones also indexes cochlear synaptopathy in ways that are similar to using a faster modulation rate on a higher-frequency tone (*Märcher-Rørsted et al., 2022*; *Ponsot et al., 2024*).

The Mongolian gerbil provides a robust model for cross-species comparisons with aging humans, due to overlapping hearing frequency ranges and experimentally tractable lifespans. Here, using young and middle-aged gerbils, we showed similar EFR decreases as seen in human listeners (*Figure 2C*). Additionally, age-related changes in the EFR were associated with confirmed CND (*Figure 2F*). CND in gerbils reached ~20% in the middle-aged 80-week group tested here, which is less than what has been observed in middle-aged humans, where CND estimates typically reach 40–50% by the fifth decade of life (*Wu et al., 2019*). However, our EFRs were still sensitive to this degree of CND, reiterating that EFRs are a sensitive metric for measuring cochlear deafferentation. Additionally, we confirmed that the gerbils used in this study did not show any changes in hearing thresholds (*Figure 2B*). Hence, they were unlikely to have strial degenerations that are known to occur in older gerbils that affect auditory thresholds (*Gratton and Schulte, 1995*). The

synapse loss patterns and EFR amplitude changes seen here in gerbils were in agreement with earlier studies using alternate rodent models (*Parthasarathy and Kujawa, 2018*; *Shaheen et al., 2015*; *Parthasarathy and Bartlett, 2011*), further confirming that age-related cochlear synapse loss is a pervasive mammalian phenomenon that can be captured using EFRs to rapid modulation frequencies (~1000 Hz).

Strong evidence links CND with altered neural coding of sounds in multiple ascending auditory stations (*Parthasarathy and Kujawa, 2018*; *Chambers et al., 2016*; *Resnik and Polley, 2021*). However, the perceptual consequences of CND on speech-in-noise abilities remain unclear (*Bramhall et al., 2019*). Evidence for overt behavioral deficits has been mixed and may depend on the specific type of task used for assessment (*Grant et al., 2020*; *Prendergast et al., 2017b*). Here, we used QuickSIN, a clinically-relevant test that we recently identified as being sensitive to changes in normal-hearing, adult populations with perceived hearing deficits (*Cancel et al., 2023*). However, tests that are further challenging in spectrotemporal complexity, such as the addition of time compression or reverberation, may further tease apart these differences (*Grant et al., 2020*; *Mepani et al., 2021*). In the current study, behavioral deficits began to emerge only at the most challenging SNR levels (*Figure 3*). However, perceptual deficits in terms of listening effort began to appear prior to behavioral changes.

Listening effort is an umbrella term that may assess multiple forms of executive functions such as cognitive resource allocation, working memory, and attention, and can be assessed by measuring isoluminous task-related changes in pupil diameter (*Peelle, 2018*; *Beatty, 1982*; *Winn et al., 2015*; *Kuchinsky et al., 2013*; *Brown et al., 2006*). The mechanisms underlying these pupillary changes are still under study (*McGinley et al., 2015*; *de Gee et al., 2020*), but are hypothesized to involve the locus coeruleus – norepinephrine system (*Joshi et al., 2016*; *Reimer et al., 2016*). Here, we observed that pupil-indexed listening effort increased in middle-aged adults, even when behavioral performance was matched with younger adults (*Figure 3E and F*). This suggests that middle-aged adults expend more effort to maintain behavioral performance, which may lead to more listening fatigue or disengagement from conversations (*Pichora-Fuller et al., 2016*; *Hornsby et al., 2016*; *Hornsby, 2013*). Potentially confounding factors impacting pupil measurement, such as the decrease of pupil dynamic range with aging (*Telek, 2018*; *Piquado et al., 2010*), participant fatigue, or task habituation (*McHaney et al., 2021*; *Brown et al., 2006*; *Tryon, 1975*), can vary between individuals for a multitude of reasons (*Ansari et al., 2021*). Here, the effects of these factors were minimized by applying trial-by-trial baseline corrections prior to analysis to match the magnitude of response between young and middle-aged adults.

Interestingly, pupil-indexed listening effort at a moderate SNR was a better predictor of behavioral performance at a more challenging SNR using two separate approaches – a Pearson's correlation and the elastic net regression model (*Figure 4B–D*). We have previously demonstrated similar results in a different test group of young adult participants (*McHaney et al., 2024*). These results suggest that the amount of effort required to maintain ceiling performance at moderate SNRs is predictive of behavioral performance at harder task difficulties. Pupillary indices at the harder task conditions may be rolling over into hyperexcitability (*McGinley et al., 2015*; *de Gee et al., 2020*) and, thus, being a poorer predictor of concomitant behavioral performance. Additionally, our elastic net regression model suggested that CND and listening effort provided complementary contributions to explaining variance on the QuickSIN task.

Even though both young and middle-aged adults had clinically normal-hearing thresholds, subtle changes within this normal range affected speech-in-noise performance (*Figure 4D*), lending support to studies suggesting that the definition of clinically 'normal' may need revision (*Hind et al., 2011*; *Hunter et al., 2020*). Our findings demonstrate a need for next-generation diagnostic measures of auditory processing that incorporate both neurophysiological encoding of the temporal elements of sound and cognitive factors associated with listening effort to better capture one's listening abilities. Future studies will directly test the link between cochlear and peripheral neural deficits and listening effort and explore further contributions of other top-down mechanisms that may influence listening effort, such as selective attention or semantic load (*Shinn-Cunningham and Best, 2008*; *McLaughlin et al., 2022*).

## Methods

### Humans

## Participants

### Recruitment

Young (*n*=38; 18–25 years of age, male = 10) and middle-aged (*n*=45; 40–55 years of age, male = 16) adult participants were recruited from the University of Pittsburgh Pitt+Me research participant registry, the University of Pittsburgh Department of Communication Science and Disorders research participant pool, and the broader community under a protocol approved by the University of Pittsburgh Institutional Review Board (IRB#21040125). Participants were compensated for their time, travel, and given an additional monetary incentive for completing all study sessions.

### Eligibility

Participant eligibility was determined during the first session of the study. Eligible participants had normal cognition determined by the MoCA≥25 (*Nasreddine et al., 2005*), normal-hearing thresholds (≤25 dB HL 250–8000 Hz), no severe tinnitus as self-reported via the THI (*Newman et al., 1996*), and LDLs>80 dB HL at 0.5, 1, and 3 kHz (*Sherlock and Formby, 2005*). Participants were not required to have specific complaints of speech perception in noise difficulties. The Beck Depression Inventory (*Beck et al., 1988*) was administered, and participants were excluded if they reported thoughts of self-harm, determined by any response to survey item nine greater than 0. Participants self-reported American English fluency. Thirty-five young (18–25 years of age, male = 10) and 37 middle-aged participants (40–55 years of age, male = 10) met all eligibility criteria and were tested further using the battery described below.

## Audiological assessment

### Otoscopy

An otoscopic examination was conducted using a Welch Allyn otoscope to examine the participant's external auditory canal, tympanic membrane, and middle ear space for excess cerumen, ear drainage, and other abnormalities. The presence of any such abnormality resulted in exclusion from the study, as these may lead to a conductive hearing loss.

### Audiogram

Hearing thresholds were collected inside a sound-attenuating booth using a MADSEN Astera[2] audiometer, Otometrics transducers (Natus Medical, Inc Middleton, WI, USA), and foam insert eartips sized to the participants' ear canal width. Tones were presented using a pulsed beat, and participants were instructed to press a response plunger if they believed that they perceived a tone being played, even if they were unsure. Extended high-frequency hearing thresholds were collected at frequencies 8, 12.5, and 16 kHz using Sennheiser circumaural headphones and Sennheiser HDA 300 transducers using the same response instructions.

### LDLs

LDLs were collected binaurally using the Otometrics transducer (Natus Medical, Inc, Middleton, WI, USA) and foam tip ear inserts. Warble tones were presented, and participants were instructed to rate the loudness on a scale of 1–7, with 7 being so loud that they would leave the room.

### DPOAEs

Outer hair cell function was assessed using DPOAEs. DPOAEs were collected from both the right and left ear individually, with a starting frequency of 500 Hz and an ending frequency of 16 kHz. The stimulus had an L1 of 75 dB SPL and an L2 of 65 dB SPL and was presented in 8 blocks of 24 sweeps in alternating polarities. Responses were collected using rubber ear inserts sized to participants' ear canal width and an ER-10D DPOAE Probe transducer (Etymotic Research Inc, Elk Grove, IL, USA).

### Noise exposure history

Participants completed the NEQ (*Johnson et al., 2017*) as a self-reported assay of annual noise exposure, accounting for both occupational and nonoccupational sources. Annual noise exposure

was expressed using $L_{Aeq8760h}$, representing the annual hourly duration of noise exposure presented in sound pressure level in dB. Calculation of the $L_{Aeq8760h}$ followed the original article (*Johnson et al., 2017*).

## OSPAN

Participants also completed the automated version of the OSPAN task (*Unsworth et al., 2005*), as a metric of working memory (*Turner and Engle, 1989*). Participants were shown simple arithmetic problems and asked to decide whether presented solutions to the problems were correct or incorrect. A letter was displayed on the screen after each math problem. Following a series of arithmetic-letter presentations, participants were required to recall the letters that were displayed in the order that they appeared. The task consisted of 15 letter sequences that spanned three to seven letters (three repetitions of each span). If a participant correctly recalled all letters from a sequence, the span length was added to their score. The maximum possible score on the OSPAN task was 75.

## Speech perception in noise

### Sentence-level speech perception in noise

Speech perception in noise was indexed using moderate-predictability sentences masked in multitalker babble at six different SNRs from the QuickSIN test (*Killion et al., 2004*). QuickSIN is a standardized measure of speech perception in noise that is commonly used in audiology clinics and is representative of a naturalistic listening environment (*Meyer et al., 2013*). Each QuickSIN test list consisted of six sentences masked in four-talker babble at the following SNR levels: 25, 20, 15, 10, 5, and 0 dB. All participants completed four test lists. Participants listened to the sentences through Sennheiser circumaural headphones. The masker was presented at 60 dB SPL, and the sound level of the target sentences was varied to obtain the required SNR level. Participants were instructed to repeat the target sentence to the best of their ability. Each target sentence contained five keywords for identification. The number of keywords identified per sentence was recorded. Then, the proportion of keywords correctly identified for each SNR across all four test lists (20 total keywords per SNR) was calculated for each participant (*Killion et al., 2004*; *Wilson et al., 2003*). Additionally, we calculated the standard clinical QuickSIN score of dB SNR loss, which reflects the lowest SNR level that an individual can accurately identify words 50% of the time. For each participant, the dB SNR loss score was calculated for each test list separately using the following equation: $25.5 - \left(\sum of\,keywords\,identified \in list\right)$ (*Killion et al., 2004*). Then, the mean dB SNR loss across all four test lists was calculated and used for analysis.

## Pupillometry

### Acquisition

Pupillary responses were recorded while participants completed the QuickSIN task. Participants were seated facing a monitor in a testing room with consistent, moderate ambient lighting. Monocular left-eye pupillary responses were recorded at a 1000 Hz sampling rate using an EyeLink 1000 Plus Desktop Mount camera and chin rest (SR Research). Nine-point eye-tracker calibration was performed prior to the start of the experiment. To start each trial, participants were required to fixate on a cross in the center of the screen for a minimum of 500 ms. This fixation criterion was applied to control for the effects of saccades, which can alter pupil diameter, and to minimize pupil foreshortening errors (*Zekveld et al., 2013*; *Winn et al., 2018*; *Koelewijn et al., 2018*). After meeting the 500 ms fixation criteria, a 100 ms 1000 Hz beep was presented to alert the participant to the start of the trial. There was a 2 s delay after the beep before the QuickSIN stimulus was presented. The background masker began 3 s before the target sentence and continued for 2 s after the target sentence. After the end of the background masker, there was a 2 s delay followed by a 100 ms 1000 Hz beep to signal the start of the verbal response period. Manual drift correction was performed at the end of each trial by the experimenter to ensure high-quality tracking of the pupil.

### Preprocessing

Pupillary data were processed in R (*R Core Team, 2022*) using the *eyelinker* package (*Barthelme, 2024*). Pupillary responses were analyzed in two windows of interest: (1) listening window, from multitalker babble onset through 5800 ms, and (2) integration window, from target sentence offset to

1000 ms prior to behavioral response period. Separately for each window of interest, data were first processed to remove noise from blinks and saccades. Any trial with more than 15% of the samples detected as saccades or blinks was removed. For the remaining trials, blinks were linearly interpolated from 60 ms before to 160 ms after the detected blinks. Saccades were linearly interpolated from 60 ms before to 60 ms after any detected saccade. The de-blinked data were then downsampled to 50 Hz. Pupillary responses were baseline-corrected and normalized on a trial-by-trial basis to account for a downward drift in baseline that can occur across a task and for individual differences in pupil dynamic range (**Winn et al., 2018**). Baseline pupil size was defined as the average pupil size in the 1000 ms period prior to the start of the window of interest ($\frac{pupil - baseline}{baseline} \times 100$). The pupillary response was then averaged across all four test lists for each SNR per participant in each window of interest. The outcome reported is percent change in pupil size from baseline.

GCAs (**Mirman, 2014**) were used to obtain a measure of the slope of the pupillary response during QuickSIN. GCA uses orthogonal polynomial time terms to model distinct functional forms of the pupillary response over time. Two GCAs were fit using a second-order orthogonal polynomial to model the interaction of age group with SNR level, separately for the listening window and the integration window. This second-order model provides three parameters to explain the pupillary response. The first is the intercept, which refers to the overall change in the pupillary response over the time-window of interest. The second is the linear term (ot1), which represents the slope of the pupillary response over time, or the rate of dilation. The third is the quadratic term (ot2), representing the curvature of the pupil response, or the change in the rate of the pupillary response over time. GCAs were conducted in R (**R Core Team, 2022**) using the *lme4* package (**Bates et al., 2015**), and p-values were estimated using the *lmerTest* package (**Kuznetsova et al., 2017**).

For the listening window, the best-fit GCA model included fixed effects of each time term (ot1, ot2), SNR (reference = 25), Group (reference = younger), and all two- and three-way interactions between SNR, Group, and time terms. The random effect structure consisted of a random slope of each time term per participant that removed the correlation between random effects, and a random slope of each time term per the interaction of participant and SNR level.

$$Pupil \sim (ot1 + ot2) * SNR * Group + (0 + ot1 + ot2 | participant) + (ot1 + ot2 | participant : SNR)$$

For the integration window, the best-fit GCA model included fixed effects of each time term (ot1, ot2), SNR (reference = 25), Group (reference = younger), and all two- and three-way interactions between SNR, Group, and time terms. The random effect structure consisted of a random slope of each time term per participant, and a random slope of each time term per the interaction of participant and SNR level.

$$Pupil \sim (ot1 + ot2) * SNR * Group + (ot1 + ot2 | participant) + (ot1 + ot2 | participant : SNR)$$

## Electrophysiology

### EFRs

EFRs were collected in a sound-attenuating booth using a BioSemi ActiveTwo EEG system while participants were seated in a recliner. Stimuli were presented using ER-3C transducers (Etymotic Research Inc, Elk Grove, IL, USA) with gold-foil tiptrodes placed in the ear canals to deliver sound stimuli and record additional channels of evoked potentials. EFRs were recorded to a 250 ms tone with a carrier frequency of 3000 Hz, amplitude modulated (AM) at 40, 110, 512, and 1024 Hz. Stimuli were presented in alternating polarity, with 500 repetitions, each at 85 dB SPL to the right ear. Each token was presented at 3.1 repetitions/s, for a period of 322 ms.

### Preprocessing

EFRs from the Fz to the ipsilateral (right) tiptrode were processed and analyzed using MATLAB v. 2022a (Mathworks Inc, Natick, MA, USA). EFRs were processed using a fourth-order Butterworth filter with a low-pass filter of 3000 Hz. The high-pass filter cutoffs used were 5 Hz, 80 Hz, 200 Hz, and 300 Hz for 40 Hz, 110 Hz, 512 Hz, and 1024 Hz AM stimuli, respectively. Fast Fourier transforms (FFTs) were performed on the averaged time domain waveforms for each participant at each AM rate, starting 10 ms after stimulus onset to exclude auditory brain stem responses (ABRs) and ending 10 ms

after stimulus offset. The maximum amplitude of the FFT peak at one of three adjacent bins (~3 Hz) around the modulation frequency of the AM rate was reported as the EFR amplitude.

## Animals

### Subjects

Fourteen young adult Mongolian gerbils aged 18–27 weeks (male = 9) and thirteen middle-aged Mongolian gerbils aged 75–82 weeks (male = 6) were used in this study. All animals are born and raised in our animal care facility and sourced from Charles River breeders. The acoustic environment within the holding facility was characterized by noise-level data logging and was periodically monitored. Data logging revealed an average noise level of 56 dB, with transients not exceeding 74 dB during regular housing conditions, and 88 dB once a week during cage changes. All animal procedures were approved by the Institutional Animal Care and Use Committee of the University of Pittsburgh (Protocol #21046600).

### Experimental setup

Experiments were performed in a double-walled acoustic chamber. Animals were placed on a water-circulated warming blanket set to 37°C with the pump placed outside the recording chamber to eliminate audio and electrical interferences. Gerbils were initially anesthetized with isoflurane gas anesthesia (4%) in an induction chamber. The animals were transferred post-induction to a manifold and maintained at 1–1.5% isoflurane. Subdermal electrodes (Ambu) were then placed on the animals' scalps for the recordings. A positive electrode was placed along the vertex. The negative electrode was placed under the ipsilateral ear along the mastoid, while the ground electrode was placed in the base of the tail. Impedances from the electrodes were always less than 1 kHz, as tested using the head-stage (RA4LI, Tucker Davis Technologies [TDT]). The average duration of isoflurane anesthesia during the electrode setup process was approximately 10 min. After placing electrodes, animals were injected with dexmedetomidine (Dexdomitor, 0.3 mg/kg subdermal) and taken off of the isoflurane. Dexmedetomidine is an alpha-adrenergic agonist that acts as a sedative and an analgesic and is known to decrease motivation, but preserve behavioral and neural responses in rodents (*Ruotsalainen et al., 1997*; *Ter-Mikaelian et al., 2013*). This helps to maintain animals in an un-anesthetized state where they still respond to pain stimuli, such as a foot pinch, but are otherwise compliant to recordings for a period of about 3 hours. The time window for the effects of isoflurane to wear off was determined empirically as 9 min, based on ABRs' waveforms and latencies, as well as the response to foot pinch stimuli. Recordings then commenced 15 min after cessation of isoflurane.

### Stimulus presentation, acquisition, and analysis

Stimuli were presented to the right ear of the animal using insert earphones (ER3C, Etymotic), which matched the stimulus presentation in humans. Stimuli presentation and acquisition were done for gerbils via LabView. The output from the insert earphones was calibrated using a Bruel Kjaer microphone and was found to be within ±6 dB for the frequency range tested. Digitized waveforms were recorded with a multichannel recording and stimulation system (RZ-6, TDT) and analyzed in MATLAB (MathWorks).

Hearing thresholds were obtained using ABRs presented to 5 ms long tone stimuli, with a 2.5 ms on- and off- ramp, at 27.1 repetitions per second. ABRs were filtered from 300 Hz to 30,000 Hz, and thresholds were determined as the minimum sound level that produced a response, as assessed using visual inspection by two blinded, trained observers.

EFRs were elicited to sinusoidally AM tones (5 ms rise/fall, 250 ms duration, 3.1 repetitions/s, alternating polarity) at a 3 kHz carrier frequency presented 30 dB above auditory thresholds obtained using ABRs at 3 kHz. The modulation frequency was systematically varied from 16 Hz to 1024 Hz AM. Responses were amplified (×10,000; TDT Medusa 4z amplifier) and filtered (0.1–3 kHz). Trials in which the response amplitude exceeded 200 μV were rejected. 250 artifact-free trials of each polarity were averaged to compute the EFR waveform. FFTs were performed on the averaged time-domain waveforms starting 10 ms after stimulus onset to exclude ABRs and ending at stimulus offset using MATLAB (MathWorks). The maximum amplitude of the FFT peak at one of three frequency bins (~3 Hz each) around the modulation frequency was recorded as the peak FFT amplitude. The FFT amplitude at the AM frequency was reported as the EFR amplitude. The noise floor of the EFR was calculated as the

average of five frequency bins (~3 Hz each) above and below the central three bins. A response was deemed significantly above the noise floor if the FFT amplitude was at least 6 dB greater than the noise floor.

## Immunohistology

Animals were transcardially perfused using a 4% paraformaldehyde solution (Sigma-Aldrich, 441244) for approximately 5 min before decapitation and isolation of the right and left cochlea. Following intra-labyrinthine perfusion with 4% paraformaldehyde, cochleas were stored in paraformaldehyde for 1 hr. Cochleae were decalcified in EDTA (Fisher Scientific, BP120500) for 3–5 days, followed by cryo-protection with sucrose (Fisher Scientific, D16500) and flash-freezing. All chemicals were of reagent grade. Cochleae were thawed prior to dissection, then dissected in PBS solution. Immunostaining was accomplished by incubation with the following primary antibodies: (1) mouse anti-CtBP2 (BD Biosciences) at 1:200, (2) mouse anti-GluA2 (Millipore) at 1:2000, (3) rabbit anti-myosin VIIa (Proteus Biosciences) at 1:200, followed by incubation with secondary antibodies coupled to Alexa Fluor in the red, green, and blue channels. Piece lengths were measured and converted to cochlear frequency using established cochlear maps (*Greenwood, 1990*) in ImageJ. Cochlear stacks were obtained at the target frequency (3 kHz) spanning the cuticular plate to the synaptic pole of ~10 hair cells (in 0.25 μm z-steps). Images were collected in a 1024×1024 raster using a high-resolution, oil-immersion objective (×60) and 1.59× digital zoom using a Nikon A1 confocal microscope. Images were denoised in NIS elements and loaded into an image-processing software platform (Imaris; Oxford Instruments), where inner hair cells were quantified based on their Myosin VIIa-stained cell bodies and CtBP2-stained nuclei. Presynaptic ribbons and postsynaptic glutamate receptor patches were counted using 3D representations of each confocal z-stack. Juxtaposed ribbons and receptor puncta constitute a synapse, and these synaptic associations were determined using IMARIS workflows that calculated and displayed the x-y projection of the voxel space (*Parthasarathy and Kujawa, 2018*; *Liberman et al., 2011*).

## Statistical analysis

### Analysis of variance

Normality of all variables was first checked visually using Q-Q plots and statistically using the Shapiro-Wilk test with alpha = 0.05. Homogeneity of variance was assessed using Levene's test. *N*-way analyses of variance (ANOVAs) were completed using R 2022.07.1 for each measure to determine statistically significant differences between groups (*Girden, 1992*). The function employed, *aov*, uses treatment contrasts in which the first baseline level is compared to each of the following levels. The number of factors was determined based on the conditions tested in each measure. Bonferroni corrections were used to control familywise error rate due to multiple comparisons.

### Correlations

Outliers were detected using Tukey's Fence with a boundary distance of k=1.5 and removed. Correlations were computed using Pearson's correlations. Degrees of freedom, *r*, and p-values were reported.

### Elastic net regression

We used a linear model with elastic net penalization/regularization (*Zou and Hastie, 2005*) to simultaneously estimate the underlying contributions of the various predictor variables measured in our studies and perform model selection. This approach has been previously validated for model selection using multidimensional data related to hearing pathologies like tinnitus and hypera-cusis (*Smith et al., 2024*). The relative strength of selection and shrinkage is controlled by the hyper-parameters $\lambda$ and $\alpha$: a higher $\lambda$ implies more stringent penalization pushing toward the null model, and $0 \leq \alpha \leq 1$ controls the degree of convexity and hence the amount of sparsity, with $\alpha = 0$ implying a Ridge regression with no variable selection. Elastic net is a regularized regression method that minimizes the negative log-likelihood with a penalty on the parameters that combines the $l_1$ (LASSO) and $l_2$ (Ridge) penalty, i.e., the elastic net penalty on the regression parameters $\beta$ can be written as $Pen\left(\beta\right) = \lambda\left(\alpha\left\|\beta\right\|_1 + (1 - \alpha)/2\left\|\beta\right\|_2^2\right)$. Elastic net regularization has several advantages over both LASSO and Ridge, as well as a simple linear model. The $l_1$ part

of the elastic net ($\|\beta\|_1$) leads to a sparse model where some of the coefficients are shrunk to exact zeroes, thereby performing an automatic model selection without the combinatorial computational complexities of a best-subset selection approach. Further, the quadratic $l_2$ part ($\|\beta\|_2^2$) encourages grouped variable selection and removes the limitation of the number of selected variables, unlike LASSO, while stabilizing the selection path. To choose the tuning parameters $\lambda$ and $\alpha$, we used a 10-fold cross-validation that minimizes the out-of-sample root mean-squared error. We used the R packages *glmnet* (*Friedman et al., 2010*) and *caret* (*Kuhn, 2008*) for training the elastic net regularizer.

## Acknowledgements

This work was supported by the National Institute on Deafness and Other Communication Disorders-National Institutes of Health Grants R21DC018882 to AP, T32DC011499 to K Kandler and B Yates (Trainee: MEZ) and F31DC020085 to JRM, and the PNC-Trees Charitable Trust (PNC to BC and AP). We thank Dr. Carl Snyderman for collaboration on the PNC-Trees grant, and Megan Hallihan, Kathryn Bergstrom, Sarah Anthony, and Shaina Wasileski for their assistance with participant recruitment and data collection. Thanks also to Dr. Simon Warkins, Katherine Helfrich, and Mike Calderon at the Center for Biological Imaging at the University of Pittsburgh, supported by NIH grant 1S10RR028478-01 for collaboration on confocal imaging, and the Clinical and Translational Science Institute at the University of Pittsburgh, supported by the NIH Clinical and Translational Science Award (CTSA) program, grant UL1 TR001857 for assistance with participant recruitment.

## Additional information

### Funding

| Funder | Grant reference number | Author |
| --- | --- | --- |
| National Institutes of Health | R21DC018882 | Aravindakshan Parthasarathy |
| National Institutes of Health | F31DC020085 | Jacie R McHaney |
| National Institutes of Health | T32DC011499 | Maggie E Zink |
| PNC Foundation | | Bharath Chandresekaran Aravindakshan Parthasarathy |

The funders had no role in study design, data collection and interpretation, or the decision to submit the work for publication.

### Author contributions

Maggie E Zink, Data curation, Formal analysis, Investigation, Visualization, Methodology, Writing – original draft, Writing – review and editing; Leslie Zhen, Data curation, Formal analysis, Supervision, Investigation, Methodology, Writing – review and editing; Jacie R McHaney, Data curation, Formal analysis, Supervision, Investigation, Visualization, Methodology, Writing – review and editing; Jennifer Klara, Kimberly Yurasits, Victoria E Cancel, Data curation, Investigation; Olivia Flemm, Data curation, Formal analysis, Investigation, Project administration; Claire Mitchell, Data curation, Formal analysis, Investigation; Jyotishka Datta, Data curation, Formal analysis, Visualization, Methodology; Bharath Chandresekaran, Supervision, Funding acquisition, Project administration, Writing – review and editing; Aravindakshan Parthasarathy, Conceptualization, Supervision, Funding acquisition, Investigation, Visualization, Methodology, Project administration, Writing – review and editing

### Author ORCIDs

Maggie E Zink ⬡ https://orcid.org/0000-0003-2018-2772

Aravindakshan Parthasarathy (iD) https://orcid.org/0000-0002-4573-8004

### Ethics

Human subjects: Informed consent and consent to publish were obtained from participants. Participants were recruited and tested under a protocol approved by the University of Pittsburgh Institutional Review Board (IRB#21040125). Participants were compensated for their time, travel, and given an additional monetary incentive for completing all study sessions.

This study was performed in strict accordance with the recommendations in the Guide for the Care and Use of Laboratory Animals of the National Institutes of Health. All animal procedures were approved by the Institutional Animal Care and Use Committee of the University of Pittsburgh (Protocol #21046600).

Reviewer #1 (Public review): https://doi.org/10.7554/eLife.102823.3.sa1
Reviewer #2 (Public review): https://doi.org/10.7554/eLife.102823.3.sa2
Author response https://doi.org/10.7554/eLife.102823.3.sa3

## Additional files

### Supplementary files

MDAR checklist

### Data availability

All data reported and analyzed in this study can be found on the Open Science Framework at http://doi.org/10.17605/OSF.IO/4BGDA.

The following dataset was generated:

| Author(s) | Year | Dataset title | Dataset URL | Database and Identifier |
|---|---|---|---|---|
| McHaney JR, Zink M, Zhen L, Klara J, Yurasits K, Cancel V, Flemm O, Mitchell C, Datta J, Chandrasekaran B, Parthasarathy A | 2024 | Increased listening effort and cochlear neural degeneration underlie behavioral deficits in speech perception in normal hearing middle-aged adults | http://doi.org/10.17605/OSF.IO/4BGDA | Open Science Framework, 10.17605/OSF.IO/4BGDA |

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

## Appendix 1

**Appendix 1—key resources table**

| Reagent type (species) or resource | Designation | Source or reference | Identifiers | Additional information |
|---|---|---|---|---|
| Genetic reagent (*Meriones unguiculatus*) | Crl:MON(Tum) | Charles River | 243 | |
| Antibody | ms(1gG1) α CtBP2 (mouse monoclonal) | BD Transduction Labs | BDB612044 | |
| Antibody | rb α MyosinVIIa (rabbit polyclonal) | Proteus Biosciences | 25-670 | |
| Antibody | ms(1gG2a) α GluA2 (mouse monoclonal) | Millipore | MAB397 | |
| Antibody | gt α ms (IgG2a) AF 488 (goat polyclonal) | Thermo Fisher | A-21131 | |
| Antibody | gt α ms (IgG1) AF 568 (goat polyclonal) | Thermo Fisher | A-21124 | |
| Antibody | dk α rb AF 647 (donkey polyclonal) | Thermo Fisher | A-31573 | |
| Chemical compound, drug | Isoflurane | Covetrus | 29405 | |
| Chemical compound, drug | Dexmedetomidine | Covetrus | 60984 | |
| Software, algorithm | Labview | National Instruments | https://www.ni.com/en-us/shop/labview.html | |
| Software, algorithm | MATLAB | MathWorks | https://www.mathworks.com/products/matlab.html | |
| Other | Eyelink 1000 Plus | SR Research | https://www.sr-research.com/eyelink-1000-plus/ | |
| Other | BioSemi Active Two EEG | BioSemi | https://www.biosemi.com/Products_ActiveTwo.htm | |
| Other | Insert earphones | Etymotic | ER-3C | |
| Other | Multi-I/O Processor- RZ6-A-P1 | TDT | https://www.tdt.com/product/rz6-multi-i-o-processor/ | |
| Other | ABR Amplifier, Medusa 4Z | TDT | https://www.tdt.com/product/medusa4z-amplifier/ | |

