## [Editor Report · eLife Assessment]

This study aims to clarify the effects of cochlear neural degeneration on auditory processing in listeners with normal audiograms (sometimes referred to as 'hidden hearing loss'). The authors provide **important** new data demonstrating associations between cochlear neural degeneration, non-invasive assays of auditory processing, and speech perception. Based on a cross-species comparison, the findings pose **compelling** evidence that cochlear synaptopathy is associated with a significant part of hearing difficulties in complex environments.

---

## [Referee Report · Reviewer #1 (Public review)]

This study is part of an ongoing effort to clarify the effects of cochlear neural degeneration (CND) on auditory processing in listeners with normal audiograms. This effort is important because ~10% of people who seek help for hearing difficulties have normal audiograms and current hearing healthcare has nothing to offer them.

The authors identify two shortcomings in previous work that they intend to fix. The first is a lack of cross-species studies that make direct comparisons between animal models in which CND can be confirmed and humans for which CND must be inferred indirectly. The second is the low sensitivity of purely perceptual measures to subtle changes in auditory processing. To fix these shortcomings, the authors measure envelope following responses (EFRs) in gerbils and humans using the same sounds, while also performing histological analysis of the gerbil cochleae, and testing speech perception while measuring pupil size in the humans.

The study begins with a comprehensive assessment of the hearing status of the human listeners. The only differences found between the young adult (YA) and middle aged (MA) groups are in thresholds at frequencies > 10 kHz and DPOAE amplitudes at frequencies > 5 kHz. The authors then present the EFR results, first for the humans and then for the gerbils, showing that amplitudes decrease more rapidly with increasing envelope frequency for MA than for YA in both species. The histological analysis of the gerbil cochleae shows that there were, on average, 20% fewer IHC-AN synapses at the 3 kHz place in MA relative to YA, and the number of synapses per IHC was correlated with the EFR amplitude at 1024 Hz.

The study then returns to the humans to report the results of the speech perception tests and pupillometry. The correct understanding of keywords decreased more rapidly with decreasing SNR in MA than in YA, with a noticeable difference at 0 dB, while pupillary slope (a proxy for listening effort) increased more rapidly with decreasing SNR for MA than for YA, with the largest differences at SNRs between 5 and 15 dB. Finally, the authors report that a linear combination of audiometric threshold, EFR amplitude at 1024 Hz, and a few measures of pupillary slope is predictive of speech perception at 0 dB SNR.

I only have two questions/concerns about the specific methodologies used:

(1) Synapse counts were made only at the 3 kHz place on the cochlea. But the EFR sounds were presented at 85 dB SPL, which means that a rather large section of the cochlea will actually be excited. Do we know how much of the EFR actually reflects AN fibers coming from the 3 kHz place? And are we sure that this is the same for gerbils and humans given the differences in cochlear geometry, head size, etc.?

[Note added after revision: the authors have added new data, references, and discussion that have answered my initial questions].

(2) Unless I misunderstood, the predictive power of the final model was not tested on held out data. The standard way to fit and test such model would be to split the data into two segments, one for training and hyperparameter optimization, and one for testing. But it seems that the only spilt was for training and hyperparameter optimization.

[Note added after revision: the authors now make it clear in their response that the modeling tells us how much of the current data can be explained but not necessary about generalization to other datasets.]

While I find the study to be generally well executed, I am left wondering what to make of it all. The purpose of the study with respect to fixing previous methodological shortcomings was clear, but exactly how fixings these shortcomings has allowed us to advance is not. I think we can be more confident than before that EFR amplitude is sensitive to CND, and we now know that measures of listening effort may also be sensitive to CND. But where is this leading us?

I think what this line of work is eventually aiming for is to develop a clinical tool that can be used to infer someone's CND profile. That seems like a worthwhile goal but getting there will require going beyond exploratory association studies. I think we're ready to start being explicit about what properties a CND inference tool would need to be practically useful. I have no idea whether the associations reported in this study are encouraging or not because I have no idea what level of inferential power is ultimately required.

[Note added after revision: the authors have added to the Discussion to put their work into a broader perspective.]

That brings me to my final comment: there is an inappropriate emphasis on statistical significance. The sample size was chosen arbitrarily. What if the sample had been half the size? Then few, if any, of the observed effects would have been significant. What if the sample had been twice the size? Then many more of the observed effects would have been significant (particularly for the pupillometry). I hope that future studies will follow a more principled approach in which relevant effect sizes are pre-specified (ideally as the strength of association that would be practically useful) and sample sizes are determined accordingly.

[Note added after revision: my intention with this comment was not to make a philosophical or nitty-gritty point about statistics. It was more of a follow on to the previous point. Because I don't know what sort of effect size is big enough to matter (for whatever purpose), I don't find the statistical significance (or lack thereof) of the effect size observed to be informative. But I don't think there is anything more that the authors can or should do in this regard.]

So, in summary, I think this study is a valuable but limited advance. The results increase my confidence that non-invasive measures can be used to infer underlying CND, but I am unsure how much closer we are to anything that is practically useful.

---

## [Referee Report · Reviewer #2 (Public review)]

Summary:

This paper addresses the bottom-up and top-down causes of hearing difficulties in middle-aged adults with clinically-normal audiograms using a cross-species approach (humans vs. gerbils, each with two age groups) mixing behavioral tests and electrophysiology.. The study is not only a follow-up of Parthasarathy et al (eLife 2020), since there are several important differences. Parthasarathy et al. (2020) only considered a group of young normal-hearing individuals with normal audiograms yet with high complaints for hearing in noisy situations. Here, this issue is considered specifically regarding aging, using a between-subject design comparing young NH and older NH individuals recruited from the general population, without additional criterion (i.e. no specifically high problems of hearing in noise). In addition, this is a cross-species approach, with the same physiological EFR measurements with the same stimuli deployed on gerbils.

This article is of very high quality. It is extremely clear, and the results show clearly a decrease of neural phase-locking to high modulation frequencies in both middle-aged humans and gerbils, compared to younger groups/cohorts. In addition, pupillometry measurements conducted during the QuickSIN task suggest increased listening efforts in middle-aged participants, and a statistical model including both EFRs and pupillometry features suggest that both factors contribute to reduced speech-in-noise intelligibility evidenced in middle-aged individuals, beyond their slight differences in audiometric thresholds (although they were clinically normal in both groups).

These provide strong support to the view that normal aging in humans leads to auditory nerve synaptic loss (cochlear neural degeneration - CND- or, put differently, cochlear synaptopathy) as well as increased listening effort, before any clearly visible audiometric deficits as defined in current clinical standards. This result is very important for the community, since we are still missing direct evidence that cochlear synaptopathy might likely underly a significant part of hearing difficulties in complex environments for listeners with normal thresholds, such as middle-aged and senior listeners. This paper shows that these difficulties can be reasonably well accounted for by this sensory disorder (CND), but also that listening effort, i.e. a top-down factor, further contributes to this problem. The methods are sound, well described and I would like to emphasize that they are presented concisely yet in a very precise manner, so that they can be understood very easily - even for a reader that is not familiar with the employed techniques. I believe this study will be of interest to a broad readership.I have some comments and questions which I think would make the paper even stronger once addressed.

Main comments:

(1) Presentation of EFR analyses / Interpretation of EFR differences found in both gerbils and humans

(a) Could you comment further on why you think you found a significant difference only at the highest mod. frequency of 1024 Hz in your study? Indeed, previous studies employing SAM or RAM tones very similar to the ones employed here were able to show age effects already at lower modulation freqs. of ~100H; e.g. there are clear age effects reported in human studies of Vasilikov et al. (2021) or Mepani et al. (2021), and also in animals (see Garrett et al. bioRxiv : https://www.biorxiv.org/content/biorxiv/early/2024/04/30/2020.06.09.142950.full.pdf)

Furthermore, some previous EEG experiments in humans that SAM tones with modulation freqs. of ~100Hz showed that EFRs do not exhibit a single peak, i.e. there are peaks not only at fm but also for the first harmonics (e.g. 2*fm or 3*fm) see e.g. Garrett et al. bioRxiv https://www.biorxiv.org/content/biorxiv/early/2024/04/30/2020.06.09.142950.full.pdf

Did you try to extract EFR strength by looking at the summed amplitude of multiple peaks (Vasilikov Hear Res. 2021), in particular for the lower modulation frequencies? (Indeed, there will be no harmonics for the higher mod. freqs).

b) How the present EFR results relate to FFR results, where effects of age are already at low carrier freqs? (e.g. Märcher-Rørsted et al., Hear. Res., 2022 for pure tones with freq < 500 Hz)Do you think it could be explained by the fact that this is not the same cochlear region, and that synapses die earlier in higher compared to lower CFs. This should be discussed.Beyond the main group effect of age, there were no negative correlations of EFRs with age in your data?

(2) Size of the effects / comparing age effects between two species:Although the size of the age effect on EFRs cannot be directly compared between humans and gerbils - the comparison remains qualitative - could you a least provide references regarding the rate of synaptic loss with aging in both humans and gerbils, so that we understand that the yNH/MA difference can be compared between the two age groups used for gerbils; it would have been critical in case of a non-significant age effect in one species.

Equalization / control of stimuli differences across the two species: For measuring EFRs, SAM stimuli were presented at 85 dB SPL for humans vs. 30 dB above detection threshold (inferred from ABRs) for gerbils - I do not think the results strongly depend on this choice, but it would be good to comment on why you did not choose also to present stimuli 30 dB above thresholds in humans.

Simulations of EFRs using functional models could have been used to understand (at least in humans) how the differences in EFRs obtained between the two groups are *quantitatively* compatible with the differences in % of remaining synaptic connections known from histopathological studies for their age range (see the approach in Märcher-Rørsted et al., Hear. Res., 2022)

(3) Synergetic effects of CND and listening effortCould you test whether there is an interaction between CNR and listening effort? (e.g. one could hypothesize that MA subjects with largest CND have also the higher listening effort)

Comments on revised version:

The authors did well to address all the points raised in my review. This paper will make an important contribution to our assessment of the sources of age-related auditory processing deficits beyond the cochlea that impair speech intelligibility.

---

## [Author Response]

The following is the authors’ response to the original reviews.

**Public Reviews:**

**Reviewer #1 (Public review):**
This study is part of an ongoing effort to clarify the effects of cochlear neural degeneration (CND) on auditory processing in listeners with normal audiograms. This effort is important because ~10% of people who seek help for hearing difficulties have normal audiograms and current hearing healthcare has nothing to offer them.The authors identify two shortcomings in previous work that they intend to fix. The first is a lack of cross-species studies that make direct comparisons between animal models in which CND can be confirmed and humans for which CND must be inferred indirectly. The second is the low sensitivity of purely perceptual measures to subtle changes in auditory processing. To fix these shortcomings, the authors measure envelope following responses (EFRs) in gerbils and humans using the same sounds, while also performing histological analysis of the gerbil cochleae, and testing speech perception while measuring pupil size in the humans.The study begins with a comprehensive assessment of the hearing status of the human listeners. The only differences found between the young adult (YA) and middle-aged (MA) groups are in thresholds at frequencies > 10 kHz and DPOAE amplitudes at frequencies > 5 kHz. The authors then present the EFR results, first for the humans and then for the gerbils, showing that amplitudes decrease more rapidly with increasing envelope frequency for MA than for YA in both species. The histological analysis of the gerbil cochleae shows that there were, on average, 20% fewer IHC-AN synapses at the 3 kHz place in MA relative to YA, and the number of synapses per IHC was correlated with the EFR amplitude at 1024 Hz.The study then returns to the humans to report the results of the speech perception tests and pupillometry. The correct understanding of keywords decreased more rapidly with decreasing SNR in MA than in YA, with a noticeable difference at 0 dB, while pupillary slope (a proxy for listening effort) increased more rapidly with decreasing SNR for MA than for YA, with the largest differences at SNRs between 5 and 15 dB. Finally, the authors report that a linear combination of audiometric threshold, EFR amplitude at 1024 Hz, and a few measures of pupillary slope is predictive of speech perception at 0 dB SNR.I only have two questions/concerns about the specific methodologies used:(1) Synapse counts were made only at the 3 kHz place on the cochlea. However, the EFR sounds were presented at 85 dB SPL, which means that a rather large section of the cochlea will actually be excited. Do we know how much of the EFR actually reflects AN fibers coming from the 3 kHz place? And are we sure that this is the same for gerbils and humans given the differences in cochlear geometry, head size, etc.?

Thank you for raising this important point. The frequency regions that contribute to the generation of EFRs, especially at the suprathreshold sound levels presented here are expected to be broad, with a greater leaning towards higher frequencies and reaching up to one octave above the center frequency. We have investigated this phenomenon in earlier published articles using both low/high pass masking noise and computational models using data from rodent models and humans (Encina-Llamas et al. 2017; Parthasarathy, Lai, and Bartlett 2016). So, the expectation here is that the EFRs reflect a wider frequency region centered at 3 kHz. The difference in cochlear activation regions between humans and gerbils for EFRs have not been systematically studied to our knowledge but given the general agreement between humans and other rodent models stated above, we expect this to be similar to gerbils as well. Additionally, all current evidence points to cochlear synapse loss with age being flat across frequencies, in contrast to cochlear synapse loss with noise which is dependent on the bandwidth of the noise exposure.

Histological evidence for this flat loss across frequencies is found in mice and human temporal bones (Parthasarathy and Kujawa 2018; Sergeyenko et al. 2013; Wu et al. 2018). We find this to be true in our gerbils as well. Author response image 1 shows the patterns of synapse loss as a function of cochlear place. We focused on synapse loss at 3 kHz to keep the analysis focused on the center frequency of the stimulus and minimize compounding errors due to averaging synapse counts across multiple frequency regions. We have now added some explanatory language in the discussion.

**Author response image 1. sa3fig1:** Cochlear synapse counts per inner hair cell (IHC) in young and middle-aged gerbils as a function of cochlear frequency.

(2) Unless I misunderstood, the predictive power of the final model was not tested on heldout data. The standard way to fit and test such a model would be to split the data into two segments, one for training and hyperparameter optimization, and one for testing. But it seems that the only split was for training and hyperparameter optimization.

The goal of the analysis in this current manuscript was inference, rather than prediction, i.e., to find the important/significant variables that contribute to speech intelligibility in noise, rather than predicting the behavioral deficit of speech performance in a yet-unforeseen sample of adults.

Additionally, we used a repeated 10-fold cross-validation approach for our model building exercise as detailed in the Elastic Net Regression section of the methods. This repeated-cross validation calculated the mean square error on a held-out fold and average it repeatedly to reduce the inherent variability of randomly choosing a validation set. The repeated 10-fold CV approach is both more stable and efficient compared to a validation set approach, or splitting the data into two segments: training and test, and provides a better estimate of the test error by utilizing more observations for training (vide Chapter 5,James et al. 2021). These predictive MSEs along with the R-squared for the final model give us a good idea of the predictive performance, as, for the linear model the R-squared is the correlation between the observed and the predicted response. Future studies with a larger sample size can facilitate having a designated test set and still have enough statistical power to perform predictive analyses.

While I find the study to be generally well executed, I am left wondering what to make of it all. The purpose of the study with respect to fixing previous methodological shortcomings was clear, but exactly how fixing these shortcomings has allowed us to advance is not. I think we can be more confident than before that EFR amplitude is sensitive to CND, and we now know that measures of listening effort may also be sensitive to CND. But where is this leading us? I think what this line of work is eventually aiming for is to develop a clinical tool that can be used to infer someone's CND profile. That seems like a worthwhile goal but getting there will require going beyond exploratory association studies. I think we're ready to start being explicit about what properties a CND inference tool would need to be practically useful. I have no idea whether the associations reported in this study are encouraging or not because I have no idea what level of inferential power is ultimately required.

Studies with CND have so far been largely inferential in humans, since currently we cannot confirm CND in vivo. Hence any measures of putative CND in humans can only be interpreted based on evidence from other animal studies. Our translational approach is partly meant to address this deficit, as mentioned in the Introduction section. By using identical stimuli, recording, acquisition and analysis parameters we hope to reduce some of the variability that may be associated with this inference between human and other animal models. Until direct measurements of CND in humans are possible, the intended goal is to provide diagnostic biomarkers that have face validity – i.e., that explain variance related to speech intelligibility deficits in this population.

We’ve added more to the discussion to state that our work demonstrates the need for next generation diagnostic measures of auditory processing that incorporate cognitive factors associated with listening effort to better capture speech in noise perceptual abilities.

That brings me to my final comment: there is an inappropriate emphasis on statistical significance. The sample size was chosen arbitrarily. What if the sample had been half the size? Then few, if any, of the observed effects would have been significant. What if the sample had been twice the size? Then many more of the observed effects would have been significant (particularly for the pupillometry). I hope that future studies will follow a more principled approach in which relevant effect sizes are pre-specified (ideally as the strength of association that would be practically useful) and sample sizes are determined accordingly.

We agree that pre-determining sample sizes is the optimal approach towards designing a study. The sample sizes here were chosen a priori based on previously published data in young adults with normal hearing thresholds (McHaney et al. 2024; Parthasarathy et al. 2020). With the lack of published literature especially for the EFRs at 1024Hz AM in middle aged adults, there are practical challenges in pre-determining the sample size (given a prefixed power and an effect size) with limited precursors to supply good estimates of the parameters (e.g., mean, s.d. for each age group for a two-sample test). We hope that this data set now shared will enable us and other researchers to conduct power analyses for successive studies that use similar metrics on this population.

Several authors, including Heinsburg and Weeks (2022) argue that post-hoc power could be “misleading and simply not informative” and encourage using other indicators of poorly powered studies such as the width of the confidence interval. Since the elastic net estimate is a non-linear and non-differentiable function of the response values—even for fixed tuning parameters—it is difficult to obtain an accurate estimate of its standard error (Tibshirani and Taylor 2012). While acknowledging the limitations of post-hoc power analyses, we performed a retrospective power calculation for our linear model with the predictors that we selected (EFR @ 1024Hz, Pupil slope for QuickSIN at selected SNRs and analyses windows, and PTA). The calculated Cohen’s effect size was 0.56, which is considered large (Cohen 2013). With this effect size, a power analysis with our sample size revealed a very high retrospective power of 0.99 with a significance level of 0.05. The minimum number of subjects needed to get 80% power with this effect size was N = 21. Hence for the final model, we are confident that our results hold true with adequate statistical power.

So, in summary, I think this study is a valuable but limited advance. The results increase my confidence that non-invasive measures can be used to infer underlying CND, but I am unsure how much closer we are to anything that is practically useful.

Thank you for your comments. We hope that this study establishes a framework for the eventual development of the next generation of objective diagnostics tests in the hearing clinic that provide insights into the underlying neurophysiology of the auditory pathway and take into effect top-down contributors such as listening effort.

**Reviewer #2 (Public review):**
Summary:This paper addresses the bottom-up and top-down causes of hearing difficulties in middleaged adults with clinically-normal audiograms using a cross-species approach (humans vs. gerbils, each with two age groups) mixing behavioral tests and electrophysiology. The study is not only a follow-up of Parthasarathy et al (eLife 2020), since there are several important differences.Parthasarathy et al. (2020) only considered a group of young normal-hearing individuals with normal audiograms yet with high complaints of hearing in noisy situations. Here, this issue is considered specifically regarding aging, using a between-subject design comparing young NH and older NH individuals recruited from the general population, without additional criterion (i.e. no specifically high problems of hearing in noise). In addition, this is a cross-species approach, with the same physiological EFR measurements with the same stimuli deployed on gerbils.This article is of very high quality. It is extremely clear, and the results show clearly a decrease of neural phase-locking to high modulation frequencies in both middle-aged humans and gerbils, compared to younger groups/cohorts. In addition, pupillometry measurements conducted during the QuickSIN task suggest increased listening efforts in middle-aged participants, and a statistical model including both EFRs and pupillometry features suggests that both factors contribute to reduced speech-in-noise intelligibility evidenced in middle-aged individuals, beyond their slight differences in audiometric thresholds (although they were clinically normal in both groups).These provide strong support to the view that normal aging in humans leads to auditory nerve synaptic loss (cochlear neural degeneration - CNR- or, put differently, cochlear synaptopathy) as well as increased listening effort, before any clearly visible audiometric deficits as defined in current clinical standards. This result is very important for the community since we are still missing direct evidence that cochlear synaptopathy might likely underlie a significant part of hearing difficulties in complex environments for listeners with normal thresholds, such as middle-aged and senior listeners. This paper shows that these difficulties can be reasonably well accounted for by this sensory disorder (CND), but also that listening effort, i.e. a top-down factor, further contributes to this problem. The methods are sound and well described and I would like to emphasize that they are presented concisely yet in a very precise manner so that they can be understood very easily - even for a reader who is not familiar with the employed techniques. I believe this study will be of interest to a broad readership.I have some comments and questions which I think would make the paper even stronger once addressed.Main comments:(1) Presentation of EFR analyses / Interpretation of EFR differences found in both gerbils and humans:a) Could the authors comment further on why they think they found a significant difference only at the highest mod. frequency of 1024 Hz in their study? Indeed, previous studies employing SAM or RAM tones very similar to the ones employed here were able to show age effects already at lower modulation freqs. of ~100H; e.g. there are clear age effects reported in human studies of Vasilikov et al. (2021) or Mepani et al. (2021), and also in animals (see Garrett et al. bioRxiv: https://www.biorxiv.org/content/biorxiv/early/2024/04/30/2020.06.09.142950.full.pdf).

Previously published studies in animal models by us and others suggests that EFRs elicited to AM rates > 700Hz are most sensitive to confirmed CND (Parthasarathy and Kujawa 2018; Shaheen, Valero, and Liberman 2015). This is likely because these AM rates fall well outside of phase-locking limits in the auditory midbrain and cortex (Joris, Schreiner, and Rees 2004), and hence represent a ‘cleaner’ signal from the auditory periphery that may not be modulated by complex excitatory/inhibitory feedback circuits present more centrally (Caspary et al. 2008). We have also demonstrated that we are able to acquire high quality EFRs at 1024Hz AM rates both in a previously published study in young normal hearing adults (McHaney et al. 2024), and in middle aged adults in the present study as seen in Fig. 1 H-J. We posit that the lack of age-related differences at the lower AM rates may be indicative of compensatory plasticity with age (central ‘gain’) that occurs with age in more central regions of the auditory pathway (Auerbach, Radziwon, and Salvi 2019; Parthasarathy and Kujawa 2018). We now expand on this in the discussion. A secondary reason for the lack of change in slower modulation rates may be the difference in stimulus between sinusoidally amplitude modulated tones used here, and the rectangular amplitude modulated tones in other studies, as discussed in response to the comment below.

Furthermore, some previous EEG experiments in humans that SAM tones with modulation freqs. of ~100Hz showed that EFRs do not exhibit a single peak, i.e. there are peaks not only at fm but also for the first harmonics (e.g. 2*fm or 3*fm) see e.g. Garrett et al. bioRxiv https://www.biorxiv.org/content/biorxiv/early/2024/04/30/2020.06.09.142950.full.pdf. Did the authors try to extract EFR strength by looking at the summed amplitude of multiple peaks (Vasilikov Hear Res. 2021), in particular for the lower modulation frequencies? (indeed, there will be no harmonics for the higher mod. freqs).

We examined peak amplitudes for the AM rate and harmonics for the 110 Hz AM condition as shown in Author response image 2. The quantified amplitudes of the first four harmonics did not differ with age (ps > .08).

Additionally, the harmonic structures obtained were also not as robust as would be expected with rectangular amplitude modulated stimuli. The choice of sinusoidal modulation may explain why. We have previously published studies systematically modulating the rise time of the envelope per cycle in amplitude modulated tones, where the individual period of the envelope is described by Env (t) = t^x^ (1-t), where t goes from 0 to 1 in one period, and where x = 0.05 represents a highly damped envelope akin to the rising envelope f a rectangular modulation, and x = 1 representing a symmetric, near-sinusoidal envelope (Parthasarathy and Bartlett 2011). The harmonic structure was much more developed in the damped envelopes compared to the symmetric envelopes and response amplitudes were also higher for the damped envelopes overall, a result also observed in Mepani et. al., 2021. Hence, we believe the rapid rise time may contribute to the harmonic structures evidenced in studies using RAM stimuli, and the absence of this rapid onset may result in reduced harmonic structures in our EFRs. Some language regarding this issue is now added to the discussion.

**Author response image 2. sa3fig2:** Harmonics analysis for the first four harmonics of envelope following responses elicited to the 110Hz AM stimulus.

b) How do the present EFR results relate to FFR results, where effects of age are already at low carrier freqs? (e.g. Märcher-Rørsted et al., Hear. Res., 2022 for pure tones with freq < 500 Hz). Do the authors think it could be explained by the fact that this is not the same cochlear region, and that synapses die earlier in higher compared to lower CFs? This should be discussed. Beyond the main group effect of age, there were no negative correlations of EFRs with age in the data?

We believe the current results are in close agreement with these studies showing deficits in pure tone phase locking with age. These tones are typically at ~300-500Hz or above, and phase locking to these tones likely involves the same or similar peripheral neural generators in the auditory nerve and brainstem. Emerging evidence also seems to suggest that TFS coding measured using pure tone phase locking is closely related to sound with amplitude modulation in the same range (Ponsot et al. 2024). Unpublished observations from our lab support this view as well. In this data set, we begin to see EFR responses at 512 Hz diverge with age, but this difference does not reach statistical significance. This may be due to specific AM frequencies selected or a lack of statistical power. Using more continuous AM frequency sweeps such as with our recently published dynamic amplitude modulated tones (Parida et al. 2024) may help resolve these AM frequency specific challenges and help us investigate changes over a broader range of AM frequencies. Ongoing studies are currently exploring this hypothesis. Some explanatory language is now presented in the discussion.

(2) Size of the effects / comparing age effects between two species:Although the size of the age effect on EFRs cannot be directly compared between humans and gerbils - the comparison remains qualitative - could the authors at least provide references regarding the rate of synaptic loss with aging in both humans and gerbils, so that we understand that the yNH/MA difference can be compared between the two age groups used for gerbils; it would have been critical in case of a non-significant age effect in one species.

Current evidence seems to suggest that humans have more synaptic loss than gerbils, though exact comparison of lifespan between the two species is challenging due to differences in slopes of growth trajectories between species. Post-mortem temporal bone studies demonstrate a ~40-50% loss of synapses in humans by the fifth decade of life. On the other hand, our gerbils in the current study showed approximately 15-20% loss. Based on our findings and previous studies, it is reasonable to assume that our gerbil data underestimate the temporal processing deficits that would be seen in humans due to CND.

We have added this information and citations to the discussion section.

Equalization/control of stimuli differences across the two species: For measuring EFRs, SAM stimuli were presented at 85 dB SPL for humans vs. 30 dB above the detection threshold (inferred from ABRs) for gerbils - I do not think the results strongly depend on this choice, but it would be good to comment on why you did not choose also to present stimuli 30 dB above thresholds in humans.

We chose to record EFRs to stimuli presented at 85 dB SPL in humans, as opposed to 30 dB SL, because 30 dB SL in humans would have corresponded to an intensity that makes EEG recordings unfeasible. The average PTA across younger and middle-aged adults was 7.51 dB HL (~19.51 dB SPL), which would have resulted in an average stimulus intensity of ~50 dB SPL at 30 dB SL. This intensity level would have been far too low to reliably record EFRs without presenting many thousands of trials. In a pilot study, we recorded EFRs at 75 dB SL, which equated to an average of 83.9 dB SPL. Thus, we chose the suprathreshold level of 85 dB SPL for the current study to obtain reliable responses with just 1000 trials.

Simulations of EFRs using functional models could have been used to understand (at least in humans) how the differences in EFRs obtained between the two groups are *quantitatively* compatible with the differences in % of remaining synaptic connections known from histopathological studies for their age range (see the approach in Märcher-Rørsted et al., Hear. Res., 2022)

We agree with the reviewer that phenomenological models would be a useful approach to examining differences between age groups and species. We have previously used the Zilany/Carney model to examine differences in EFRs with age in rats (Parthasarathy, Lai, and Bartlett 2016). It is unclear if such models will directly translate to responses form gerbils. However, this is a subject of ongoing study in our lab.

(3) Synergetic effects of CND and listening effort:Could you test whether there is an interaction between CND and listening effort? (e.g. one could hypothesize that MA subjects with the largest CND have also higher listening effort).

We have previously reported that EFRs and listening effort are not linearly related (McHaney et al. 2024). We found the same to be largely true in the current study as well. We ran correlations between EFR amplitudes at 1024 Hz and listening effort at each SNR level in the listening and integrations windows. We did not observe any significant relationships between EFRs at 1024 Hz and listening effort in the listening window (all *ps* > .05). In the integration window, we did see a significant correlation between listening effort at SNR 5 and EFRs at 1024 Hz, which was significant after correcting for multiple comparisons (*r* = -.42, *p*-adj = .021). However, we chose to not report these multiple oneto-one correlations in the current study and instead opted for the elastic net regression analysis to better understand the multifactorial contributions to speech-in-noise abilities. These results also do not preclude non-linear relationships between listening effort and EFRs which may be present based on emerging results (Bramhall, Buran, and McMillan 2025), and will be explored in future studies.

**Recommendations for the authors:**

**Reviewer #1 (Recommendations for the authors):**
A few more minor comments/questions:(1) How old were the YA gerbils on average? 18 weeks, or 19 weeks, or 22 weeks?

Young gerbils were on average 22 weeks. We have updated the manuscript accordingly.

(2) "Gerbils share the same hearing frequency range as humans" is misleading; the gerbil hearing range extends to much higher frequencies.

We have revised the statement to say: “The hearing range of gerbils largely overlaps with that of humans, making them an ideal animal model for direct comparison in crossspecies studies.”

(3) The writing contains more than a few typos and grammatical errors.

We have completed a thorough revision to correct for grammatical and typographical errors.

(4) Suggesting that correlation and linear modelling are "independent" methods is misleading since they are both measuring linear associations. A better word would be "different".

Thank you for this suggestion. We have rephrased the sentence as “two separate approaches”

(5) The phrase "Our results reveal perceptual deficits ... driven by CND" in the abstract is too strong. Correlation is not causation.

We have revised this phrase to say they “are associated with CND.”

**Reviewer #2 (Recommendations for the authors):**
More general comments:(1) Recruitment criterion related to hearing-in-noise difficulties:If I understood correctly, the middle-aged participants recruited for this study do not have specific hearing in noise difficulties, some could, as with 10% in the general population, but they were not recruited using this criterion. If this is correct, this should be stated explicitly, as it constitutes an important methodological choice and a difference with your eLife 2020 study. If you were to use this specific recruitment criterion for both groups here, what differences would you expect?

Our participants were not required to have specific complaints of speech perception in noise challenges to be eligible for this study. We included middle-aged adults here, as opposed to only younger adults as in Parthasarathy et al. (2020), with the assumption that middle-aged adults were likely to have some cochlear synapse loss and individual variability in the degree of synapse loss based on post-mortem data from human temporal bones. We have recently published studies identifying the specific clinical populations of patients with self-perceived hearing loss, including those adults who have received assessments for auditory processing disorders (Cancel et al. 2023). Ongoing studies in the lab are aimed at recruiting from this population.

It is striking here that the QuickSIN test does not exhibit the same variability at low SNRS here as with the digits-in-noise used in your eLife 2020 study. Why would QuickSIN more appropriate than the Digits-in-noise test? Would you expect the same results with the Digits-in-noise test?

Our 2020 eLife study investigated the effects of TFS coding in multi-talker speech intelligibility. TFS coding is specifically hypothesized to be related to multi-talker speech, compared to broadband maskers. The digits test was appropriate in that context as the ‘masker’ there was two competing speakers also speaking digits. In this study, we wanted to test the effects of CND on speech in noise perception using clinically relevant speech in noise tests. The Digits test is devoid of linguistic context and is essentially closed set (participants know that only a digit will be presented). However, QuickSIN consists of open set sentences of moderate context, making it closer to real world listening situations. Additionally, we recently published pupillometry recorded in response to QuickSIN in young adults (McHaney et al. 2024) and identified QuickSIN as a promising screening tool for self-perceived hearing difficulties (Cancel et al. 2023). These factors informed our choice of using QuickSIN in the current study.

(2) Why is the increase in listening effort interpreted as an increase in gain? please clarify (p10, 1st paragraph; [these data suggest a decrease in peripheral neural coding, with a concomitant increase in central auditory activity or 'gain'])

In the above referenced paragraph, we were discussing the increase in 40 Hz AM rate EFRs in middle-aged adults as an increase in central gain. We have revised parts of this paragraph to better communicate that we were discussing the EFRs and not listening effort: “We observed decreases in EFRs at modulation rates that were selective to the auditory periphery (i.e., 1024 Hz) in middle-aged adults, while EFRs primarily generated from the central auditory structures were not different from those in younger adults (Fig. 1K). These data suggest that middle-aged adults exhibited an increase in central auditory activity, or ‘gain’, in the presence of decreased peripheral neural coding. The perceptual consequences of this gain are unclear, but our findings align with emerging evidence suggesting that gain is associated with selective deficits in speech-in-noise abilities”

(3) Further discussion on the relationship/differences between markers EFR marker of CND (this study) and MEMR marker of CND(Bharadwaj et al., 2022) is needed.

We now make mention of other candidate markers of CND (ABR wave I and MEMRs) in the discussion and expand on why we chose the EFR.

(4) Further analyses and discussion would be needed to be related to extended high-freq thresholds:Did you test for a potential correlation of your EFR marker of CND with extended high-freq. thresholds ? (could be paralleling the amount of CND in these individuals) Why won't you also consider measuring extended HF in Gerbils?

We acknowledge that there is increasing evidence to suggest extended high frequency thresholds may be an early marker for hidden hearing loss/CND. We have examined an additional correlation for extended high frequency pure tone averages (8k-16k Hz) with EFR amplitudes at 1024 Hz AM rate, which revealed a significant relationship (*r* = -.43, *p* < .001). However, we opted to exclude this analysis from our current study as we wanted to reduce reporting on several one-to-one correlations. Therefore, we chose the elastic net regression model to examine individual contributions to speech in noise abilities. EHF thresholds were included in the elastic net regression models, but were not found to be significant upon accounting for individual differences in PTA.

Additionally, our electrophysiological experimental paradigm was not designed with the consideration of extended high frequencies—we used ER3C transducers which are not optimal for frequencies above ~6kHz. Future studies could use transducers such as the ER2 or free field speakers to examine the influence of extended high frequencies on the EFRs and measure high frequency thresholds in gerbils.

Minor Comments:(1) Abstract: repetition of 'later in life' in the first two sentences - please reformulate.

We have revised the first two sentences to state: “Middle-age is a critical period of rapid changes in brain function that presents an opportunity for early diagnostics and intervention for neurodegenerative conditions later in life. Hearing loss is one such early indicator linked to many comorbidities in older age.”

(2) Sentence on page 3 [However, these behavioral readouts may minimize subliminal changes in perception that are reflected in listening effort but not in accuracies (26-28)] is not clear.

We’ve added a sentence just after that states: “Specifically, two individuals may show similar accuracies on a listening task, but one individual may need to exert substantially more listening effort to achieve the same accuracy as the other.”

(3) The second paragraph of page 11 should go to a methods (model) section, not to the discussion.

We have now moved a portion of this paragraph to the Elastic Net Regression subsection of the Statistical Analysis in the Methods.

(4) Please checks references: references 13 and 25 are identical.

Fixed

References

Auerbach, Benjamin D., Kelly Radziwon, and Richard Salvi. 2019. “Testing the Central Gain Model: Loudness Growth Correlates with Central Auditory Gain Enhancement in a Rodent Model of Hyperacusis.” Neuroscience 407:93–107. https://doi.org/10.1016/j.neuroscience.2018.09.036.

Bramhall, Naomi F., Brad N. Buran, and Garnett P. McMillan. 2025. “Associations Between Physiological Indicators of Cochlear Deafferentation and Listening Effort in Military Veterans with Normal Audiograms.” Hearing Research, April, 109263. https://doi.org/10.1016/j.heares.2025.109263.

Cancel, Victoria E., Jacie R. McHaney, Virginia Milne, Catherine Palmer, and Aravindakshan Parthasarathy. 2023. “A Data-Driven Approach to Identify a Rapid Screener for Auditory Processing Disorder Testing Referrals in Adults.” Scientific Reports 13 (1): 13636. https://doi.org/10.1038/s41598-023-40645-0.

Caspary, D. M., L. Ling, J. G. Turner, and L. F. Hughes. 2008. “Inhibitory Neurotransmission, Plasticity and Aging in the Mammalian Central Auditory System.” Journal of Experimental Biology 211 (11): 1781–91. https://doi.org/10.1242/jeb.013581.

Cohen, Jacob. 2013. Statistical Power Analysis for the Behavioral Sciences. 2nd ed. New York: Routledge. https://doi.org/10.4324/9780203771587.

Encina-Llamas, Gerard, Aravindakshan Parthasarathy, James Michael Harte, Torsten Dau, Sharon G. Kujawa, Barbara Shinn-Cunningham, and Bastian Epp. 2017. “Hidden Hearing Loss with Envelope Following Responses (EFRs): The off-Frequency Problem: 40th MidWinter Meeting of the Association for Research in Otolaryngology.” In .

James, Gareth, Daniela Witten, Trevor Hastie, and Robert Tibshirani. 2021. An Introduction to Statistical Learning: With Applications in R. Springer Texts in Statistics. New York, NY: Springer US. https://doi.org/10.1007/978-1-0716-1418-1.

Joris, P. X., C. E. Schreiner, and A. Rees. 2004. “Neural Processing of Amplitude-Modulated Sounds.” Physiological Reviews 84 (2): 541–77. https://doi.org/10.1152/physrev.00029.2003.

McHaney, Jacie R., Kenneth E. Hancock, Daniel B. Polley, and Aravindakshan Parthasarathy. 2024. “Sensory Representations and Pupil-Indexed Listening Effort Provide Complementary Contributions to Multi-Talker Speech Intelligibility.” Scientific Reports 14 (1): 30882. https://doi.org/10.1038/s41598-024-81673-8.

Parida, Satyabrata, Kimberly Yurasits, Victoria E. Cancel, Maggie E. Zink, Claire Mitchell, Meredith C. Ziliak, Audrey V. Harrison, Edward L. Bartlett, and Aravindakshan Parthasarathy. 2024. “Rapid and Objective Assessment of Auditory Temporal Processing Using Dynamic Amplitude-Modulated Stimuli.” Communications Biology 7 (1): 1–10. https://doi.org/10.1038/s42003-024-07187-1.

Parthasarathy, A., and E. L. Bartlett. 2011. “Age-Related Auditory Deficits in Temporal Processing in F-344 Rats.” Neuroscience 192:619–30. https://doi.org/10.1016/j.neuroscience.2011.06.042.

Parthasarathy, A., J. Lai, and E. L. Bartlett. 2016. “Age-Related Changes in Processing Simultaneous Amplitude Modulated Sounds Assessed Using Envelope Following Responses.” Jaro-Journal of the Association for Research in Otolaryngology 17 (2): 119–32. https://doi.org/10.1007/s10162-016-0554-z.

Parthasarathy, A., Kenneth E Hancock, Kara Bennett, Victor DeGruttola, and Daniel B Polley. 2020. “Bottom-up and Top-down Neural Signatures of Disordered Multi-Talker Speech Perception in Adults with Normal Hearing.” Edited by Barbara G Shinn-Cunningham, Huan Luo, Fan-Gang Zeng, and Christian Lorenzi. eLife 9 (January):e51419. https://doi.org/10.7554/eLife.51419.

Parthasarathy, Aravindakshan, and Sharon G. Kujawa. 2018. “Synaptopathy in the Aging Cochlea: Characterizing Early-Neural Deficits in Auditory Temporal Envelope Processing.” The Journal of Neuroscience. https://doi.org/10.1523/jneurosci.324017.2018.

Ponsot, Emmanuel, Pauline Devolder, Ingeborg Dhooge, and Sarah Verhulst. 2024. “AgeRelated Decline in Neural Phase-Locking to Envelope and Temporal Fine Structure Revealed by Frequency Following Responses: A Potential Signature of Cochlear Synaptopathy Impairing Speech Intelligibility.” bioRxiv. https://doi.org/10.1101/2024.12.11.628010.

Sergeyenko, Yevgeniya, Kumud Lall, M. Charles Liberman, and Sharon G. Kujawa. 2013. “Age-Related Cochlear Synaptopathy: An Early-Onset Contributor to Auditory Functional Decline.” Journal of Neuroscience 33 (34): 13686–94. https://doi.org/10.1523/jneurosci.1783-13.2013.

Shaheen, L. A., M. D. Valero, and M. C. Liberman. 2015. “Towards a Diagnosis of Cochlear Neuropathy with Envelope Following Responses.” J Assoc Res Otolaryngol. https://doi.org/10.1007/s10162-015-0539-3.

Tibshirani, Ryan J., and Jonathan Taylor. 2012. “Degrees of Freedom in Lasso Problems.” The Annals of Statistics 40 (2): 1198–1232. https://doi.org/10.1214/12-AOS1003.

Wu, P. Z., L. D. Liberman, K. Bennett, V. de Gruttola, J. T. O’Malley, and M. C. Liberman. 2018. “Primary Neural Degeneration in the Human Cochlea: Evidence for Hidden Hearing Loss in the Aging Ear.” Neuroscience. https://doi.org/10.1016/j.neuroscience.2018.07.053.